# LARGE LANGUAGE MODELS TO ENHANCE BAYESIAN OPTIMIZATION

**Tennison Liu,*** **Nicolás Astorga,*** **Nabeel Seedat & Mihaela van der Schaar**
DAMTP, University of Cambridge
Cambirdge, UK
`{tl522,nja46}@cam.ac.uk`

## ABSTRACT

Bayesian optimization (BO) is a powerful approach for optimizing complex and expensive-to-evaluate black-box functions. Its importance is underscored in many applications, notably including hyperparameter tuning, but its efficacy depends on efficiently balancing exploration and exploitation. While there has been substantial progress in BO methods, striking this balance remains a delicate process. In this light, we present `LLAMBO`, a novel approach that integrates the capabilities of Large Language Models (LLM) within BO. At a high level, we frame the BO problem in natural language, enabling LLMs to iteratively *propose* and *evaluate* promising solutions conditioned on historical evaluations. More specifically, we explore how combining contextual understanding, few-shot learning proficiency, and domain knowledge of LLMs can improve model-based BO. Our findings illustrate that `LLAMBO` is effective at zero-shot warmstarting, and enhances surrogate modeling and candidate sampling, especially in the early stages of search when observations are sparse. Our approach is performed in context and does not require LLM finetuning. Additionally, it is modular by design, allowing individual components to be integrated into existing BO frameworks, or function cohesively as an end-to-end method. We empirically validate `LLAMBO`'s efficacy on the problem of hyperparameter tuning, highlighting strong empirical performance across a range of diverse benchmarks, proprietary, and synthetic tasks.

## 1 INTRODUCTION

**Black-box optimization.** Expensive black-box functions are common in many disciplines and applications including robotics [1, 2], experimental design [3], drug discovery [4], interface design [5] and, in machine learning, hyperparameter tuning [6, 7, 8]. *Bayesian optimization* (BO) is an efficient model-based approach for globally optimizing these functions [9, 10]. BO's effectiveness lies in its ability to operate based on a limited set of observations without the need for direct access to the objective function or its gradients. It does so by using observed data to learn a *surrogate model* to approximate the black-box function and a *candidate point sampler* to iteratively propose potentially good points. In each trial, the acquisition function selects the proposed point with the highest utility, based on surrogate evaluations. This chosen point undergoes evaluation, and the cycle continues.

**Challenges of search efficiency.** For BO, the name of the game is efficient search, but this efficiency largely depends on the quality of the surrogate model and candidate point sampler to quickly identify high-potential regions [11]. Given that BO is designated for scenarios with limited observations, constructing an accurate ▶ **surrogate model** with sparse observations is inherently challenging. Additionally, the model can be sensitive to misspecification, and even slight misrepresentations of the model can introduce undesired bias, skewing the ▶ **sampling of potential solutions** [12]. A further challenge arises when considering the integration of ▶ **prior knowledge**, especially in effectively transferring knowledge about correlations in the optimization space to new tasks.

At the core, these challenges pertain to accurately *learning* the objective function and effectively *generating* candidate solutions with limited data. This scenario is typically framed as the *few-shot*

---

*Equal contribution

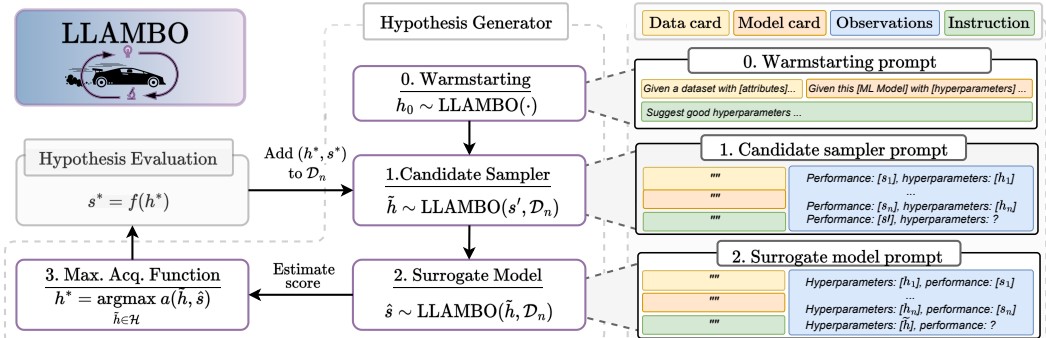

Figure 1: **Overview of LLAMBO.** In order: LLAMBO can initialize BO through ▶ **zero-shot warm-starting**, ▶ efficiently **sample candidate points** from high-potential regions given past observations and problem description, and ▶ evaluate these candidate points via a **surrogate model**.

*setting*, a context that demands swift learning and generalization from very few examples [13]. Interestingly, such challenges of the few-shot paradigm align with the proficiencies of Large Language Models (LLM). Contemporary LLMs, which have been pre-trained on Internet-scale data, showcase an exceptional capacity to generalize from sparse data, enabling them to excel in few-shot prediction, generation [14, 15, 16, 17, 18], and contextual understanding [19, 20]. They achieve this remarkable sample-efficient performance, in part, by exploiting encoded priors [21, 22].

**Key considerations.** This study examines the potential of extending the capabilities of LLMs beyond standard natural language tasks to enhance model-based BO. Our approach is grounded in representing BO components using natural language, introducing novel methods to effectively capture LLM's distinct strengths. This exploration gives rise to two key questions: **[Q1]** *Can LLMs, with their encoded knowledge and few-shot learning abilities, enhance key elements of BO, including the surrogate model and candidate point sampler?* **[Q2]** *How effectively can LLM-augmented BO components operate as a cohesive, end-to-end pipeline?* In answering these questions, we chose *hyperparameter tuning* (HPT) as our initial area of investigation. This is for two main reasons: firstly, the extensive knowledge potentially acquired by LLMs about HPT during pretraining, coupled with its relatively low-dimensional nature, makes it an ideal test bed to probe the applications of LLMs within BO. Secondly, HPT is practically important and a core enabler in many applications.

**Contributions.** We present LLAMBO, a novel approach for integrating the capabilities of LLMs into BO. To understand the performance gains from this integration, we execute a systematic investigation, exploring the aforementioned questions. Our primary contributions are:

- We propose LLAMBO, a novel approach to enhance components of model-based BO with LLMs,
- We systematically investigate the enhancements of LLAMBO throughout the BO pipeline, showcasing significant improvements to ▶ **zero-shot warmstarting**, the ▶ **efficacy of the surrogate model**, and the ▶ **efficiency of candidate sampling**,
- We empirically investigated the end-to-end performance of LLAMBO for hyperparameter tuning, demonstrating strong performance on diverse benchmarks.

## 2    LLAMBO: LLMs TO ENHANCE BO

Figure 1 illustrates the LLAMBO framework. Fundamentally, our methodology translates different components in the BO pipeline into natural language. This allows the LLM to iteratively suggest and evaluate solutions, informed both by the BO problem description and search history.

### 2.1    THE INTEGRATION OF LLMs INTO BO

**Preliminaries.** To aid with exposition, we introduce the following notation. Let us consider an objective function, $f : \mathcal{H} \to \mathcal{S}$, where $h \in \mathcal{H} \subseteq \mathbb{R}^d$ is the $d$-dimensional input, and $\mathcal{S} \in \mathbb{R}$ is the output space. We aim to find $h^* \in \mathcal{H}$ that minimizes this objective function:

$$h^* = \arg\min_{h \in \mathcal{H}} f(h)$$

where $f$ is a costly black-box function without accessible gradient information. To overcome these limitations, BO employs a surrogate model to approximate $f$ and a candidate sampler to generate

$h \in \mathcal{H}$. In general terms, a surrogate model can be viewed as a machine learning (ML) method producing the *predictive distribution* of $s$ given $h$ and some observed data $\mathcal{D}_n = \{(h_i, s_i)\}_{i=1}^n$:

$$p(s|h; \mathcal{D}_n) = \int_\Theta p(s|h, \theta; \mathcal{D}_n) p(\theta|h; \mathcal{D}_n) \, d\theta$$

Here, the marginalization is over $\theta$, a latent variable that captures the underlying structure between $s$ and $h$. Specifically, $p(\theta|\mathcal{D}_n, h) \propto p(\mathcal{D}_n|\theta, h)p(\theta)$ describes the posterior distribution after observing some data, with $p(\theta)$ being the prior knowledge of this underlying structure. The candidate point sampler can be viewed similarly, as generating samples from a posterior distribution: $p(h|\mathcal{D}_n) = \int_\Theta p(h|\theta, \mathcal{D}_n)p(\theta|\mathcal{D}_n) \, d\theta$. These priors, $p(\theta)$, can play a significant role, especially given the typically sparse observations in BO [23].[1] However, in practice, many BO applications adopt non-informative priors, potentially missing out on valuable domain-specific knowledge. The challenge lies not just in the inclusion of prior knowledge, but also in accurately learning the associated predictive distribution with ML methods, especially in settings with limited observations.

**Synergy of LLMs and BO.** In this light, LLMs can offer significant enhancements due to the following capabilities: **(1) Prior knowledge:** Recently, [25] explained LLM in-context learning (ICL) as performing implicit Bayesian inference [14]. This raises the interesting prospect of using ICL to tap into an LLM's encoded knowledge for BO. In this framework, $p(\theta)$ represents the priors related to the optimization problem and domain-specific correlations absorbed through pretraining [15, 26]. **(2) ICL:** Learning generalizable models given only limited observations is highly challenging. LLMs have demonstrated the capacity to generalize from a few in-context examples, an ability that can directly complement BO's needs for sample-efficient exploration [14, 27, 28]. **(3) Contextual understanding:** LLMs are adept at processing contextual information, especially via natural language [29]. This offers a versatile interface to incorporate *meta-features* about optimization tasks, search spaces, and auxiliary details that can improve search performance.

**Operationalizing this synergy.** Despite these hypothesized advantages of LLMs, effectively capitalizing on them in an *iterative optimization* framework like BO is challenging. Recently, [30] explored the use of LLM-based BO for molecules. While this work primarily focused on the surrogate model, we introduce novel methods for LLM enhancement of multiple components of BO and conduct a systematic investigation to understand the performance gains offered by this integration. Specifically, we employ ICL to enhance three key components of BO (Figure 1):

- **Warmstarting:** Warmstarting initializes the optimization process with a pre-identified set of $n$ points, denoted as $\{h_i\}_{i=1}^n$, which are evaluated first to build up a meaningful representation of $f$. We propose a strategy to identify promising initializations through *zero-shot* prompting.
- **Sampling candidates:** The sampling process proposes points $\{\tilde{h}_k\}_{k=1}^K$ that are considered for future evaluations. Drawing inspiration from TPE [6], we propose a mechanism to conditionally sample candidates based on a target objective value $s'$: $\tilde{h}_k \sim p(h|s'; \mathcal{D}_n)$. Here, we employ ICL by providing the optimization history as few-shot examples.
- **Surrogate modeling:** The surrogate model, denoted as $p(s|h; \mathcal{D}_n)$, is an approximation of $f$ and is trained using $\mathcal{D}_n$. Specifically, we introduce two methods, leveraging ICL on the optimization history: a *discriminative* approach that produces regression estimates with uncertainty, and a *generative* approach that scores via binary classification.

**Overview of investigation.** Having outlined our approach for leveraging LLMs in BO, we now describe the structure of our investigative study. The framework is presented below, and centres around the two aforementioned questions **[Q1-2]**. We begin by analyzing each component in isolation while keeping other factors consistent whenever possible. We conclude our study with an assessment of LLAMBO's performance as an end-to-end BO method.

| Section | Method | Goal and Method | Q's |
|---------|--------|-----------------|-----|
| Section 4 | Warmstarting | *Enhancing optimization with warmstarting from LLM prior* | **[Q1]** |
| Section 5 | Surrogate model | *Improving quality of surrogate model in few-shot settings through ICL* | **[Q1]** |
| Section 6 | Candidate sampling | *Conditional sampling of high-potential points for desired $s^*$ via ICL* | **[Q1]** |
| Section 7 | End-to-end BO | *Augmenting end-to-end BO performance* | **[Q1][Q2]** |

[1]It is worth mentioning the different approaches to encode priors: Gaussian Processes [7] embed prior distributions over functions $p(f)$, while Bayesian NNs use prior distributions over weights $p(w)$ [24].

**Experimental setup.** We conduct our investigations using 74 tasks extracted from Bayesmark and HPOBench [31, 32] and OpenAI's GPT-3.5 Language Model (see Appendix D for detailed experimental procedures). While it is important to recognize that the choice of LLM can substantially influence the results of optimization, we note that the overarching methodology and fundamental insights covered in this work are broadly applicable *beyond* the specifics of any single LLM.

## 2.2 BO Prompt Design

The proposed integrations are realized through structured natural language queries to the LLM. While the specifics of each query differ (e.g. for surrogate modeling and sampling), they are constructed from three essential elements. For the complete prompts, please refer to Appendix C.1.

- **Problem description.** This includes information of the input space $\mathcal{H}$, the output space $\mathcal{S}$, and the objective function $f$. Specifically for HPT, this entails a `<MODEL CARD>`, describing the ML model being optimized ($f$), the hyperparameters ($\mathcal{H}$), and the scoring metric ($\mathcal{S}$). We also include a `<DATA CARD>` containing dataset attributes.
- **Optimization history.** The history contains the sequence of points and scores observed during the optimization process, captured in $\mathcal{D}_n$. The observed points are provided as few-shot examples for ICL of the surrogate model and candidate point sampler.
- **Task instructions.** For each component under consideration (e.g. surrogate model), we include task-specific instructions on desired inference and guidelines on the format of the response.

## 3 Related Works

**Bayesian optimization.** At its core, BO relies on probabilistic modeling. One widely adopted technique is the Gaussian Processes (GP) due to their flexibility and analytical tractability [7, 33]. Recent works have sought to enhance their expressiveness through deep kernel GP [34, 35] and manifold GPs [36]. On another front, NN-based and tree-based surrogates have been considered, particularly due to their flexibility in high-dimensional or hierarchical optimization problems [24, 37, 38, 39]. Tree-structured Parzen Estimator (TPE) is an alternate approach based on the generative surrogate model $p(h|s)$ [6, 40, 41]. Recent trends have also leaned towards Transformers as surrogate models [42, 43]. BO is commonly used to optimize expensive, black-box functions, including in robotics and experimental design [1, 2, 3] and most prominently for autoML [40, 44, 45, 46].

**Transfer learning for BO.** Recent research in BO has explored transfer learning to improve optimization across similar domains. Prominent among these are multitask GPs, designed to optimize several related black-box functions by leveraging common structures or patterns across tasks [47, 48, 49]. Other approaches have sought to transfer learnings from previously optimized functions to new functions [50, 51]. However, these approaches only consider inductive transfer over a fixed search space, i.e. all tasks share the same search space. More recently, [42] introduced a pretrained Transformer for meta-learning across tasks with different search spaces. In our work, we explore a *lightweight* alternative by harnessing prior knowledge contained in generalist LLMs, which does not require dedicated pretraining and structured results collected from related optimization problems.

**LLMs and optimization.** Recent works have explored the use of LLMs for optimization tasks, notably for prompt optimization [52, 53, 54] and as genetic search operators in evolutionary algorithms [55, 56, 57]. Of particular note is the research that delves into LLM for BO of molecules in [30], which primarily focused on surrogate modeling. In contrast, we introduce novel methods for LLM enhancement of multiple components of model-based BO and conduct a systematic investigation to understand the performance gains offered by this integration.

## 4 Warmstarting the BO Process

**Motivation.** We start by analyzing whether LLMs can transfer prior knowledge about an optimization problem through warmstarting. While warmstarting can accelerate convergence by supplying more insightful initial points, conventional approaches require prior results collected from similar optimization problems [58, 59]. This data collection can be resource-intensive and might not be feasible for certain applications. In contrast, we explore the use of LLMs for warmstarting as a more efficient and lightweight alternative, allowing the acquisition of warmstarting points without explicitly requiring data collection from related problems.

**Method.** `LLAMBO` employs *zero-shot prompting* to sample points for warmstarting. We explore three distinct settings, each providing different levels of information about the optimization problem. ▶ **No context**: the LLM is prompted to recommend good initial hyperparameters for a given ML model, but no dataset details are provided; ▶ **Partial context**: provides *meta-features* about the dataset through the `<DATA CARD>`, including the number of samples, features, the type of features (categorical *vs* continuous), and the learning task (e.g. classification); ▶ **Full context**: further augments the `<DATA CARD>` with information on marginal distributions, inter-feature correlations, and feature-label correlations.

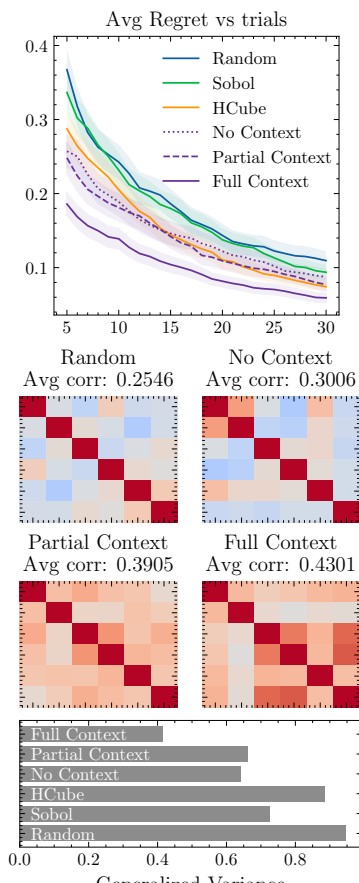

Figure 2: **Warmstarting. (Top)** average regret, **(Middle)** correlation of sampled points, **(Bottom)** diversity of initial points sampled with different methods.

**Experimental setup.** To evaluate the impact of warmstarting, we employ two widely adopted BO methods: Gaussian Processes (GP) [7] and Tree Parzen Estimator (TPE) [6]. We compare these against random initialization techniques, namely Random, Sobol, and Latin Hypercube (HCube) sampling. Each search begins with 5 initialization points and proceeds for 25 trials, and we report average results over ten seeded searches. Our evaluation metrics focus on two aspects: *search performance* and the *diversity* of the initialization points. To assess search performance, we adopt the normalized regret metric, defined as $\min_{h \in \mathcal{H}_t}(f(h) - s^*_{min})/(s^*_{max} - s^*_{min})$, where $\mathcal{H}_t$ denotes the points chosen up to trial $t$, and $s^*_{min}$ and $s^*_{max}$ represent the best and worse scores, respectively [60]. To assess diversity, we use the generalized variance: $\det(\Sigma)$, with $\Sigma$ being the covariance matrix of the hyperparameters.

**Empirical insights. (1) Performance:** Figure 2 (Top) visualizes the average regret across all tasks. We begin our analysis with a *sanity check*—namely, warmstarting using **no context** surpasses the performance of random initialization techniques. This verifies that our LLM possesses a basic knowledge of generalizable correlations (independent of specific problems) between hyperparameters. Interestingly, we observe that providing additional information about the dataset improves the search performance when warmstarting for both **partial context** and **full context**. This is particularly prominent in the early stages of the search (i.e. trials < 5). However, these initial gains are maintained as the search progresses. **(1a) Correlations:** To explore deeper, we compute the correlation matrix of sampled warmstarting points depicted in (Middle) (with further analysis in Appendix E). Our findings reveal that the points recommended by the LLM exhibit considerably greater correlations between hyperparameters compared to those from random initialization. More strikingly, the correlation matrices computed for different tasks reveal different correlation structures, suggesting that the LLM is dynamically adjusting its suggestions to different optimization problems. **(2) Diversity:** A closer look at the diversity of warmstarting points in (Bottom) reveals that their generalized variance is typically lower than that of randomly initialized points. This trend aligns with our expectations: higher correlations often lead to a decreased determinant of the covariance matrix due to 'redundant' information. Since random initialization methods sample each hyperparameter independently, they exhibit lower correlation levels, resulting in higher diversity.

> 💡 *Warmstart initialization via zero-shot prompting is an efficient strategy to transfer knowledge about correlations in the optimization landscape, enhancing search performance.*

## 5 SURROGATE MODELING

**Motivation.** Surrogate modeling, a core component of BO, aims to learn accurate representations of complex functions using only a limited set of evaluations. The efficacy of these models depends on their capacity to generalize and make accurate predictions from sparse observations. Recent

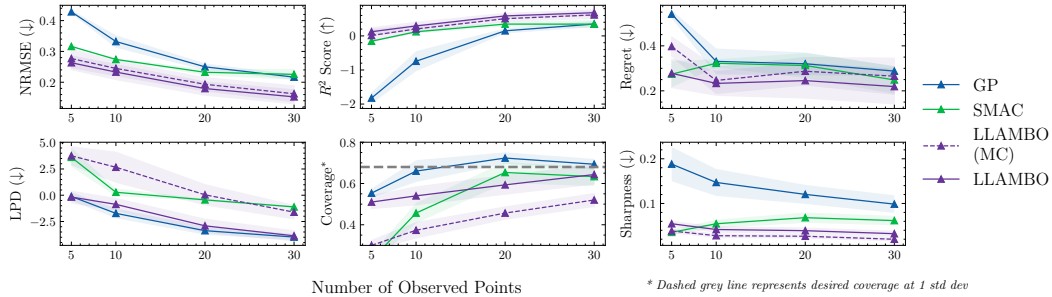

Figure 3: **Discriminative surrogate models. (Top)** prediction performance measured in NRMSE and $R^2$, and regret; **(Bottom)** uncertainty calibration evaluated using LPD, coverage, and sharpness.

studies have underscored LLM's remarkable ability to perform few-shot learning [25, 61]. Building on this, we propose two tailored approaches to surrogate modeling via ICL: **(1)** a *discriminative* approach to predict the mean and uncertainty of the objective value of a given candidate point, i.e. $p(s|h; \mathcal{D}_n)$ in Section 5.1; and **(2)** a *generative* approach that scores each point based on the probability that its objective value is better than some performance threshold $\tau$, i.e. $p(s \leq \tau | h; \mathcal{D}_n)$ in Appendix B. These represent two distinct approaches, with **(1)** framing surrogate modeling as a regression problem, while **(2)** views surrogate modeling as probabilistic binary classification.

## 5.1 DISCRIMINATIVE SURROGATE MODEL

One of the main approaches to surrogate modeling involves learning the conditional probability of the output $s$ given the input $h$ using data $\mathcal{D}_n$, expressed as $p(s|h; \mathcal{D}_n)$—a *discriminative* approach. An effective surrogate model should produce an accurate mean prediction of the objective function's central tendencies, and well-calibrated uncertainty estimates to balance exploration and exploitation.

**Method.** We serialize the observed *optimization trajectory* into natural text. For example, with $h_i$ as an RF's hyperparameters and $s_i$ the accuracy, the serialization would read: *"max_depth is 15, min_samples_split is 0.5,. . ., accuracy is 0.9"* [62, 63]. These text representations, for all $n$ observed samples, are concatenated into few-shot examples, symbolized as $\mathcal{D}_n^{\text{nl}}$. Here, we use the superscript $^{\text{nl}}$ to mean representations of observations in natural text. Together with the problem description and query example $h_k^{\text{nl}}$, they form the input to the LLM. For each query, the LLM outputs a response: $(\hat{s}_k, p(\hat{s}_k))$, denoting the predicted score and associated probability, respectively: $(\hat{s}_k, p(\hat{s}_k)) = \text{LLAMBO}(h_k^{\text{nl}}, \mathcal{D}_n^{\text{nl}})$. To obtain probabilistic estimates, this prediction step is repeated $K$ times, from which we compute the empirical mean and standard deviation. This Monte Carlo-based approach is termed `LLAMBO (MC)`, and mirrors the method proposed in [30].

Our empirical observations revealed that the MC implementation often achieved suboptimal calibration of uncertainty estimates. After further explorations, we found the sensitivity to the ordering of in-context examples as a likely cause. As LLMs process inputs in a left-to-right manner, the predictions are sensitive to permutations within the prompt [64, 65]. To enhance robustness, we introduce a shuffling mechanism that randomly permutes the few-shot examples within $\mathcal{D}_n^{\text{nl}}$, which is combined with MC sampling. We acknowledge that while this approach is not grounded in principled probabilistic reasoning—similar to the popular SMAC method [8]—it can be an effective technique to obtain probabilistic estimates. This improved method is hereby referred to as `LLAMBO`.

**Experimental setup.** We compare `LLAMBO` against GP and SMAC, two established surrogate models. We evaluate these probabilistic discriminative models via *prediction performance* and *uncertainty calibration*. For performance metrics, we use NRMSE ($\downarrow$) and $R^2$ ($\uparrow$). Calibration is assessed using the scoring rule, log predictive density (LPD) ($\downarrow$), empirical coverage (where the desired coverage for 1 standard deviation, assuming Gaussianity, is $\approx 0.68$), and sharpness ($\downarrow$) [66]. We also include normalized regret of the point acquired using expected improvement (EI) [10]. Our goal is to assess the surrogate model's efficacy when a different number of evaluations are available ($n$). We evaluate each task when $n \in [5, 10, 20, 30]$, and we test predictions against 20 unseen points.

**Empirical insights. (1) Prediction performance:** Figure 3 (Top) plots the NRMSE and $R^2$ against the number of observed samples. `LLAMBO` consistently outperforms in prediction across all sam-

ple counts, particularly with fewer observed samples. Moving on, we examine normalized regret: notably, all methods show increased regret at $n = 5$, this reflects greater uncertainty across unexplored regions, leading to heightened levels of exploration (and higher regret). For $n > 5$, LLAMBO attains lower regret, demonstrating better exploitation than other methods. **(2) Uncertainty quantification:** (Bottom) assesses uncertainty quantification. We find that, in this aspect, GPs, with their probabilistic grounding, produce the best uncertainty estimates, particularly in LPD and empirical coverage. GPs maintain good coverage even with a low number of samples, while LLAMBO only approaches similar performances as $n$ increases. In this regard, our approach exhibits performance more similar to SMAC, a frequentist method that also makes use of empirical variance. Interestingly, we note that the sharpness of uncertainty intervals for GPs remains consistently higher, while in LLAMBO, the sharpness decreases as the coverage improves. This is likely due to the better prediction performance, enabling the predictions to be more confident (lower sharpness) while achieving improved empirical coverage. **(3) LLAMBO *vs* LLAMBO (MC)** The purely MC-driven approach exhibits subpar uncertainty calibration, evident through worse LPD and coverage metrics. Coupled with low sharpness values, this suggests the predictions are overly confident, tending to underestimate uncertainty. We also observe that LLAMBO consistently achieves better prediction performance.

As such, empirical evidence supports that permuting few-shot examples, while straightforward in implementation, improves both uncertainty quantification and prediction performance, both critical aspects of balancing exploration and exploitation. **(4) Role of prior knowledge** Lastly, we investigate the importance of prior knowledge to LLAMBO's few-shot performances. To this end, we introduce an ablation setting LLAMBO (UnInf) where the problem description (containing the <DATA CARD> and <MODEL CARD>) are omitted, and the hyperparameter names are substituted with "$X_i$". Figure 4 reveals better prediction performance and calibration when compared to the uninformative ablation. This reveals the crucial role of prior knowledge in enhancing surrogate modeling, especially in few-shot settings [25, 63].

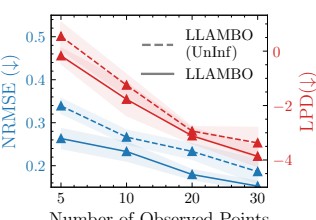

Figure 4: **Ablation.** LLAMBO (UnInf) omits problem description and hyperparameter names.

> 💡 *Discriminative surrogate models implemented through ICL can produce effective regression estimates with uncertainty, although there is a tradeoff of stronger prediction performance with worse calibration than probabilistic methods. The LLM's encoded prior is crucial to improving the efficacy of such surrogate models.*

## 6 SAMPLING OF CANDIDATE POINTS

**Motivation.** The sampling of candidate points is another crucial component of BO, as high-potential points can speed up convergence to the optimal solution. In this context, we present a novel mechanism to conditionally generate candidate points based on desired objective values through ICL.

**Method.** Our proposed sampling mechanism draws inspiration from TPE. While TPE focuses on sampling candidate points, denoted as $\tilde{h}_m$, from 'good' regions in the search space (i.e. $\tilde{h}_m \sim l(h) = p(h|s \leq \tau; \mathcal{D}_n)$), we sample from regions of high potential by directly conditioning on a desired objective value $s'$: $\tilde{h}_m \sim p(h|s'; \mathcal{D}_n)$. This distinction is fundamental as it allows us to target specific objective values, something TPE's binary categorization cannot achieve. The few-shot generation capabilities of LLMs are crucial here, as learning such a conditional generator through conventional means poses significant challenges due to the limited number of observations.

We define the desired objective value using the equation: $s' = s_{min} - \alpha \times (s_{max} - s_{min})$, where $s_{max}$ and $s_{min}$ are the worst and best objective values observed up until that point. Intuitively, $s'$ is defined relative to the best objective value, with the difference proportional to the observed variability in $s$. The exact value is controlled by $\alpha$, the *exploration* hyperparameter. A positive $\alpha$ sets $s'$ to improve over $s_{min}$. Here, we are essentially extrapolating, which cannot be achieved through conventional TPE. Conversely, a negative $\alpha$ (i.e. $-1 \leq \alpha < 0$) results in a more conservative target value that is within the observed objective value range. To operationalize this, we implement $p(h|s'; \mathcal{D}_n)$ through ICL. We generate $M$ candidate points independently, i.e. $\tilde{h}_k \sim \text{LLAMBO}(s', \mathcal{D}_n^{n1})$, after which, we

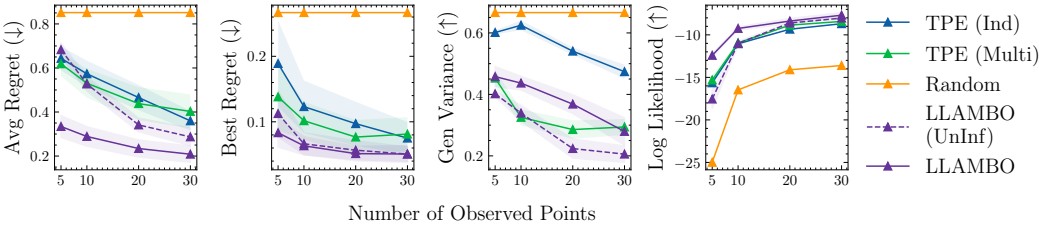

Figure 6: **Candidate point sampling.** Quality is evaluated using average regret and best regret. Diversity is assessed using generalized variance and log-likelihood.

select the point that maximizes the acquisition function as the point to evaluate next. Thus, our approach, like TPE, uses a sampling-based approximation to optimize the acquisition function.

**Experimental setup.** We compare our proposed sampler against TPE (Ind), TPE (Multi), and random sampling (Random). As before, we also include ablation of our method LLAMBO (UnInf), which omits problem description and hyperparameter names. Our analysis examines two aspects: *candidate point quality* and *diversity*. To evaluate quality, we compute the average regret ($\downarrow$) and best regret ($\downarrow$) among the $M$ sampled points [60]. For assessing diversity, we use generalized variance ($\uparrow$) to evaluate the spread of candidate points and log-likelihood ($\uparrow$) to assess the probability of candidate points being sampled from observed points. We start by investigating the effect of $\alpha$ on sampling performance. Then, following the experimental procedure outlined previously, we evaluate sampling performance when a different number of observations are available.

**Empirical insights. (1) Effect of $\alpha$:** In Figure 5, we observe that as $\alpha$ increases from $-0.5$ to $0$, both average regret and best regret improves. However, as $\alpha$ increases beyond $0$, the average regret increases as the candidate points are increasingly sampled from beyond the observed distribution, compromising the reliability of these points. Interestingly, the optimal best regret emerges at $\alpha=0.01$, hinting at our mechanism's ability to extrapolate from observed distributions. The generalized variance decreases with increasing $\alpha$, this is reasonable as the candidate points are sampled from smaller regions in the search space. Similarly, the log-likelihood decreases as $\alpha$ increases, as the points are increasingly sampled away from observed points. To confirm that this is indeed the case, we visually examine t-SNE projections of sampled points, localizing them against good (top 20% of samples) and bad points [67]. We note that when $\alpha=-0.2$, the candidate points cover a similar region as good points, but when $\alpha=0.01$, the sampled points are observed outside the regions of good points. **(2) Quality:** Figure 6 compares the quality of our sampled points against baselines, with our method set at $\alpha=-0.2$. We observe that LLAMBO consistently

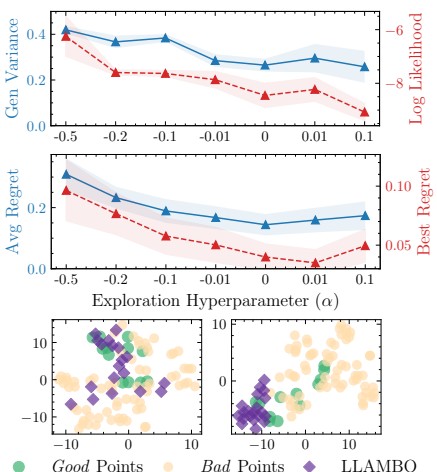

Figure 5: **(Top)** quality and diversity of points sampled with different $\alpha$. **(Bottom)** projection of sampled points at $\alpha=-0.2$ and $\alpha=0.01$.

achieves the lowest average and best regret as $n$ varies, but is especially notable at $n=5$. This gain is also present when compared against the ablation, suggesting the crucial role of prior knowledge in proposing high-potential candidates. **(3) Diversity:** An examination of generalized variance reveals that TPE (Ind) proposes more diverse points. In contrast, the spread of LLAMBO-sampled points is similar to that achieved by TPE (Multi). This is reasonable, as both LLAMBO and TPE (Multi) model correlations, while TPE (Ind) models each dimension independently (and higher correlation decreases generalized variance). Furthermore, the log-likelihood of LLAMBO proposed points are the highest, indicating that they are more plausible given the observed points.

💡 *Sampling candidate points by direct conditioning on desired target value can generate high-quality points, although this can sacrifice diversity among sampled points. The $\alpha$ exploration hyperparameter allows balancing of this trade-off.*

## 7 END-TO-END DEMONSTRATION OF LLAMBO

**Motivation.** Having examined the integration of LLMs into key components of BO, we now holistically evaluate the performance of LLAMBO as an end-to-end BO algorithm. Here, we instantiate LLAMBO with our discriminative surrogate model, as this is the most classic form of surrogate modeling.

**Experimental setup.** We evaluate BO performance on 25 tasks extracted from Bayesmark [31], a continuous HPT benchmark. Here, a *task* is a dataset-ML model pair, and we consider all 5 included datasets and 5 ML models. Additionally, we introduce 3 proprietary and 2 synthetic datasets into the benchmark—these are datasets for which the LLM would not have seen during pretraining, and thus

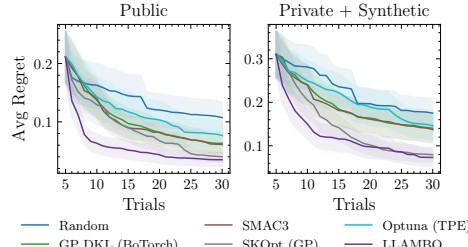

Figure 7: **End-to-end performance of LLAMBO. (Left)** average regret on public datasets and **(Right)** private and synthetic datasets evaluated on Bayesmark.

serve to check for any memorization concerns. This results in a total of 50 HPT tasks, where for each task, we executed 5 seeded searches, each with 25 trials. **Baselines.** We compare LLAMBO against 4 established baselines commonly used in production: GP-DKL [34], SKOpt (GP) [68], Optuna (TPE) [41], and SMAC3 (RF) [8]. To ensure a fair comparison, we do not use warmstarting and initialize all methods with the same set of 5 randomly sampled points in each run. We describe complete experimental details in Appendix D.

**Empirical insights. (1) Performance:** Figure 7 shows the average regrets across all HPT tasks on both public Bayesmark datasets, and private and synthetic datasets. We note that in both settings, LLAMBO achieves the best tuning performance. Additionally, we observe that, consistent with prior findings, LLAMBO excels in earlier stages of the search, when fewer observations are available. **(2) Additional results:** In the interest of space, we include additional results in Appendix E. Specifically, we ▶ evaluate LLAMBO with our generative surrogate model; ▶ compare against additional baselines on Bayesmark; ▶ evaluate BO performance on 24 additional tasks from HPOBench [32]; ▶ and report individual task search results (by task metric, average regret, and average rank).

> 💡 *LLAMBO performs effectively as an end-to-end pipeline, exhibiting sample-efficient search. Its modularity further enables individual components to be integrated into existing frameworks.*

## 8 DISCUSSIONS

In summary, we introduced LLAMBO, a novel framework that integrated LLM capabilities to enhance model-based BO. Our approach introduced three specific enhancements: ▶ **zero-shot warmstarting** to initialize search, generative and discriminative ▶ **surrogate models** of the objective function via ICL, and a ▶ **candidate point sampler** that can conditionally generate for specific target values. Our investigative study on the problem of HPT uncovered performance improvements across all three integrations, which was especially notable when fewer samples were available. Additionally, we found that LLAMBO to be an effective stand-alone BO method, exemplified through superior performance on diverse benchmarks.

**Limitations & future works.** While LLAMBO does not perform any finetuning, performing inference through LLMs incurs a much larger computational footprint than traditional BO algorithms. Our findings indicated that LLAMBO trades off this computational complexity for improved sample efficiency, an especially desirable property in black-box optimization tasks. This suggests the potential fusion of LLAMBO with more computationally efficient methods. For instance, deploying LLAMBO in earlier stages of the search, or only leveraging an individual component to complement existing BO frameworks. Additionally, while we have demonstrated the potential for integrating LLM in BO with GPT-3.5, it is important to recognize the choice of LLMs can significantly influence optimization results. A promising future direction involves benchmarking various LLMs, to understand their strengths and limitations in different BO problem settings. Our study has primarily focused on HPT tasks, which are relatively low-dimensional. However, a notable area for future research is the expansion LLAMBO's application to higher-dimensional BO tasks with more complex search spaces, such as neural architecture search and robotic control [39, 69].

ETHICS AND REPRODUCIBILITY STATEMENTS

**Ethics.** In this work, we evaluate both public benchmarks and private datasets. The private datasets are *de-identified* and used following the guidance of the respective data providers. We follow recommendations to use the Azure OpenAI service when using GPT models, where via the agreement we ensure the medical data is not sent for human review or stored, hence respecting the guidelines given by the dataset providers.

**Reproducibility.** Experimental investigations are described in Sections 4 to 7 with further details of the method, experimental setup, and datasets included in Appendix Appendix D. We provide the code to reproduce our results at `https://github.com/tennisonliu/LLAMBO` and the wider lab repository `https://github.com/vanderschaarlab/LLAMBO`.

ACKNOWLEDGMENTS

We thank the anonymous ICLR reviewers, members of the van der Schaar lab, and Andrew Rashbass for many insightful comments and suggestions. TL would like to thank AstraZeneca for their sponsorship and support. NA thanks W.D. Armstrong Trust for their support. NS is funded by the Cystic Fibrosis Trust. This work was supported by Microsoft's Accelerate Foundation Models Academic Research initiative.

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

# A  BAYESIAN OPTIMIZATION BACKGROUND

## A.1  PRELIMINARIES

Consider an objective function: $f : \mathcal{H} \rightarrow \mathcal{S}$, where $h \in \mathcal{H}$ is the input (which could be $\mathbb{R}^d$ for $d$-dimensional input) and $\mathcal{S} \in \mathbb{R}$ is the output space. We aim to find $h^* \in \mathcal{H}$ that minimizes the objective function:

$$h^* = \underset{h \in \mathcal{H}}{\arg\min} \, f(h)$$

However, $f$ is assumed to be a black-box function and evaluations of $f$ might be expensive. To address these limitations, BO is a promising method that constructs a surrogate model to approximate $f$ and iteratively proposes potential points. In more detail, the core components are:

1. **Surrogate model**: BO methods typically construct a surrogate approximation of $f$ using available samples. We denote the surrogate model $p(s|h; \mathcal{D}_n)$, where $\mathcal{D}_n := \{(h_i, s_i)\}_{i=1}^n$ are the $n$ observed input-output pairs. Some commonly used models include the Gaussian Process (GP) [7], and random forests (SMAC) [8]. An alternative approach is the Tree Parzen Estimator (TPE) [6] which uses two hierarchical processes $l(x)$ and $g(x)$ to model input distributions when the objective function is above or below a specified quantile $\tau$: $p(h|s; \mathcal{D}_n) = l(h)$ if $h < \tau$, else $g(h)$.

2. **Candidate point sampler:** The sampler proposes a set of candidate points $\tilde{\mathcal{H}} := \{\tilde{h}_i\}_{i=1}^K$ to query next. We denote the sampler $p(h|\mathcal{D}_n)$, where each candidate point is sampled independently $\tilde{h}_i \sim p(h|\mathcal{D}_n)$. For GPs, the candidate points are typically randomly sampled but then further optimized directly using the acquisition function. In SMAC, candidate points are sampled using a combination of random search and local search in the good regions found by the random forest. For TPE, the candidate points are sampled directly from the density of "good" points, $g(h)$.

3. **Acquisition function:** The acquisition function $a : \mathcal{H} \rightarrow \mathbb{R}$ scores and selects the candidate points using the surrogate model. One of the most popular acquisition functions is expected improvement (EI): $a(h) = \mathbb{E}[\max(p(s|h) - f(h_{best}), 0)]$, where $f(h_{best})$ is the value of the best observation so far [10]. The point that maximizes the acquisition function is selected as the next sample to evaluate $h = \arg\max_{\tilde{h} \in \{\tilde{\mathcal{H}}\}} a(\tilde{h})$. Other popular acquisition functions include the probability of improvement [9] and upper confidence bound [70].

The BO process thus operates by first updating the surrogate model with existing data, and then sampling a set of promising candidate points. Using the surrogate model, the acquisition function scores each candidate point and selects the best point for evaluation. The new point and observed value are appended to available observations, and the cycle continues.

# B  GENERATIVE SURROGATE MODEL [† MOVED DOWN FROM MAIN PAPER]

**Method.** An alternative approach to surrogate modelling is to learn the *generative process* of inputs given the output, represented as $p(h|s; \mathcal{D}_n)$. This approach is exemplified in TPE-based methods [6], which constructs two hierarchical processes formulated as $p(h|s; \mathcal{D}_n) = l(h)$ if $s \leq \tau$ else $g(h)$. Here $l(h)$ is the generative model of *good* points, and $g(h)$ is the model for *bad* points. In this context, good points refer to those with score $s \leq \tau$, where $\tau$ defines the threshold of the top $\gamma$ quantile of observed $s$. Conversely, bad points correspond to scores $s > \tau$, or the bottom $1 - \gamma$ quantile of results. TPE evaluates each point using the acquisition function $a(h) \propto l(h)/g(h)$, which intuitively scores each point based on their likelihood of being good over bad. At first glance, it appears challenging to learn the two densities $l(h)$ and $g(h)$ directly with an LLM.

**Density ratio estimation.** To address this, we employ Bayes' rule, transforming the density ratio estimation into probabilistic *binary classification* [71]. In introducing TPE, [6] showed that the EI of each point is defined as such:

$$EI(h) \propto \left(\gamma + \frac{g(h)}{l(h)}(1 - \gamma)\right)^{-1}$$

In other words, to maximize EI, we would like $h$ that has a high probability under $l(h)$ and a low probability under $g(h)$. By simple application of Bayes rule, we can rewrite this as:

$$
\begin{aligned}
\left(\gamma + \frac{g(h)}{l(h)}(1-\gamma)\right)^{-1} &= \left(\gamma + \frac{p(h|s > \tau)}{p(h|s \leq \tau)}(1-\gamma)\right)^{-1} \\
&= \left(\gamma + \frac{p(s > \tau|h)p(s \leq \tau)}{p(s \leq \tau|h)p(s > \tau)}(1-\gamma)\right)^{-1} \\
&= \left(\gamma + \frac{p(s > \tau|h)p(s \leq \tau)}{p(s \leq \tau|h)}\right)^{-1} \\
&= \left(\frac{\gamma}{p(s \leq \tau|h)}\right)^{-1} \\
&= \gamma^{-1}p(s \leq \tau|h)
\end{aligned}
$$

Thus, the expression is equivalently rewritten as $\gamma^{-1}p(s \leq \tau|h)$, which is proportional to $p(s \leq \tau|h)$, i.e. the probability of $h$ belonging to the *good* points. Intuitively, this gives us a scoring function that scores each sample according to the probability of producing good objective values. Using this reformulation, we can now estimate the score $a(h)$ through ICL, by obtaining the probabilistic classification $p(s \leq \tau|h)$. We recategorize the observed samples, such that $z_i = \mathbb{1}(s_i \leq \tau)$, meaning the label is 1 if the performance exceeds the desired threshold. As before, we transform the observed samples to text $\mathcal{D}_n^{\mathtt{nl}} := \{(h_i^{\mathtt{nl}}, z_i^{\mathtt{nl}})\}^n$, obtaining $K$ predictions from the LLM, $(\hat{z}_k, p(\hat{z}_k)) = \mathtt{LLAMBO}(h_k^{\mathtt{nl}}, \mathcal{D}_n^{\mathtt{nl}})$, and computing empirical average to estimate $a(h)$.

**Experimental setup.** We compare the performance of the proposed generative surrogate model vs two variants of TPE: TPE (Ind) which models each dimension independently, and TPE (Multi) which models joint multivariate densities [44]. We evaluate the *scoring performance* and *regret* attained by these surrogate models. To assess scoring performance, we report the correlation between estimated scores $a(h)$ with ground-truth scores. Additionally, we calculate regret with respect to the point that was assigned the highest score. As before, we evaluate each task when $n \in [5, 10, 20, 30]$, and we test predictions against 20 unseen samples.

**Empirical insights. (1) Scoring performance.** Figure 8 (Top) visualizes the correlation between surrogate model predicted scores and ground truth objective values. We observe that $\mathtt{LLAMBO}$ achieves notably higher correlations, especially at $n$=5, where the TPE variants generated scores that are only weakly correlated with the ground truth performance. We note that as $n$ increases, the correlation of baseline scores improves, but $\mathtt{LLAMBO}$ is still superior. **(2) Regret** (Bottom) examines the regret of the

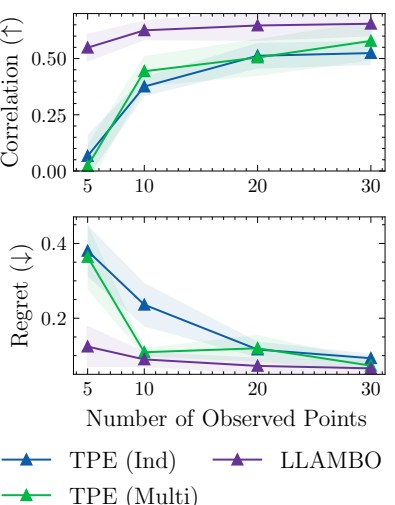

Figure 8: **Generative surrogate model. (Top)** score correlation and **(Bottom)** regret.

point picked from the 20 available points. We find that while all methods show higher regret at low sample sizes when levels of exploration are higher, $\mathtt{LLAMBO}$ quickly identifies good regions in the search space, leading to lower regret as $n$ increases.

> 💡 *Generative surrogate modeling via ICL predicts scores that are more highly correlated with ground-truth scores, leading to better identification of high-potential points.*

## C  PROMPT DESIGNS

In this section, we aim to shed light on the prompt design process. We start by supplying the complete prompts used in $\mathtt{LLAMBO}$ in Appendix C.1, before performing an ablation study in Appendix C.2 to analyze the effects of different components of the prompt.

## C.1 COMPLETE PROMPTS

In this section, we include the complete prompts used for each component of BO, including:

1. Zero-shot prompts for warmstarting with ▶ **No Context** (Figure 9); ▶ **Partial Context** (Figure 10); and ▶ **Full Context** (Figure 11).
2. ICL prompts for ▶ **discriminative surrogate model** (Figure 13) and ▶ **generative surrogate model** (Figure 14).
3. ICL prompts for target value conditioned **candidate sampling** in Figure 15.

Note that in all figures, {} is used to indicate placeholders. Each prompt is constructed with the four key components highlighted in Figure 1 and Section 2.2:

- <Model Card> describing the ML model being optimized;
- <Data Card> providing information about the dataset;
- Instructions task-specific guidelines on the format and requirements of the response;
- Observations of current optimization trajectory.

> You are assisting me with automated machine learning using {model}. I'm exploring a subset of hyperparameters detailed as: {configurations, type, and ranges}. Please suggest {number of recommendations} diverse yet effective configurations to initiate a Bayesian Optimization process for hyperparameter tuning. You mustn't include "None" in the configurations. Your response should include only a list of dictionaries, where each dictionary describes one recommended configuration. Do not enumerate the dictionaries.

Figure 9: Prompt for warmstarting with **No Context**.

> You are assisting me with automated machine learning using {model} for a {task} task. The {task} performance is measured using {metric}. The dataset has {number of samples} samples with {number of features} total features, of which {number of continuous features} are numerical and {number of categorical variables} are categorical. Class distribution is {class distribution}. I'm exploring a subset of hyperparameters detailed as: {configuration and type}. Please suggest {number of recommendations} diverse yet effective configurations to initiate a Bayesian Optimization process for hyperparameter tuning. You mustn't include 'None' in the configurations. Your response should include only a list of dictionaries, where each dictionary describes one recommended configuration. Do not enumerate the dictionaries.

Figure 10: Prompt for warmstarting with **Partial Context**.

> You are assisting me with automated machine learning using {model} for a {task} task. The {task} performance is measured using {metric}. The dataset has {number of samples} samples with {number of features} total features, of which {number of continuous features} are numerical and {number of categorical features} are categorical. Class distribution is {class distribution}. {statistical information} I'm exploring a subset of hyperparameters detailed as: {configuration and type}. Please suggest {number of recommendation} diverse yet effective configurations to initiate a Bayesian Optimization process for hyperparameter tuning. You mustn't include 'None' in the configurations. Your response should include only a list of dictionaries, where each dictionary describes one recommended configuration. Do not enumerate the dictionaries.

Figure 11: Prompt for warmstarting with **Full Context**.

> Considering one-hot encoding for categorical features the total amount input's features of the random forest is {total of features after one hot encoding}. We are standarizing numerical values to have mean 0 and std 1. The Skewness of each feature is {skewness of each feature}. The number of features that have strong correlation (defined as > 0.5 or <-0.5) with the target feature is {number of "strong correlations" with the target feature}. Of the {number of pairwise feature relationships} pairwise feature relationships, {number of "strongly correlated" pairwise features} pairs of features are strongly correlated (>0.5, <-0.5).

Figure 12: Prompt of {statistical information} used in **Full Context**.

The following are examples of the performance of a {model} measured in {metric} and the corresponding model hyperparameter configurations. The model is evaluated on a tabular {task} task containing {number of classes} classes. The tabular dataset contains {number of samples} samples and {number of features} features ({number of categorical features} categorical, {number of continuous features} numerical). Your response should only contain the predicted accuracy in the format ## performance ##.
Hyperparameter configuration: {configuration 1}
Performance: {performance 1}
...
Hyperparameter configuration: {configuration n}
Performance: {performance n}
Hyperparameter configuration: {configuration to predict performance}
Performance:

Figure 13: Prompt for **discriminative surrogate model**.

The following are examples of the performance of a {model} measured in {metric} and the corresponding model hyperparameter configurations. The model is evaluated on a tabular {task} task containing {number of classes} classes. The tabular dataset contains {number of samples} samples and {number of features} features ({number of categorical features} categorical, {number of continuous features} numerical). The performance classification is 1 if the configuration is in the best-performing 25.0% of all configurations, and 0 otherwise. Your response should only contain the predicted performance classification in the format ## performance classification ##.
Hyperparameter configuration: {configuration 1}
Classification: {classification 1}
...
Hyperparameter configuration: {configuration n}
Classification: {classification n}
Hyperparameter configuration: {configuration to classify}
Classification:

Figure 14: Prompt for **generative surrogate model**.

The following are examples of the performance of a {model} measured in {metric} and the corresponding model hyperparameter configurations. The model is evaluated on a tabular {task} task containing {number of classes} classes. The tabular dataset contains {number of samples} samples and {number of features} features ({number of categorical features} categorical, {number of continuous features} numerical). The allowable ranges for the hyperparameters are: {configuration and type}. Recommend a configuration that can achieve the target performance of {target score}. Do not recommend values at the minimum or maximum of allowable range, do not recommend rounded values. Recommend values with the highest possible precision, as requested by the allowed ranges. Your response must only contain the predicted configuration, in the format ## configuration ##.
Performance: {performance 1}
Hyperparameter configuration: configuration 1
...
Performance: {performance n}
Hyperparameter configuration: {configuration n}
Performance: {performance used to sample configuration}
Hyperparameter configuration:

Figure 15: Prompt for **candidate sampling**.

## C.2 Ablation Study

Our prompt design aligns with the framework outlined in Section 2.2, encompassing three essential components: (1) the description of the optimization problem, (2) the optimization history, and (3) explicit task instructions. To evaluate the influence of each component on performance, we conducted an ablation study with the following configurations:

- **LLAMBO**: Represents the standard LLAMBO configuration employed in our experiments.
- **LLAMBO [No context]**: This variant assesses the impact of the optimization problem description (refer to component (1)) on performance. Specifically, it omits metadata about the underlying dataset from the prompts (see Figures 16 and 17 for details).
- **LLAMBO [No instructions]**: In this setting, we exclude additional, non-formatting-related instructions from the prompts (refer to component (3)). This includes removing guidelines in the candidate point samplers regarding the types of points to be sampled. As such, the instructions that are maintained are purely to ensure adherence to the required format for regex processing (see Figure 18).

---

The following are examples of the performance of a {model} measured in {metric} and the corresponding model hyperparameter configurations. ~~The model is evaluated on a tabular {task} task containing {number of classes} classes. The tabular dataset contains {number of samples} samples and {number of features} features ({number of categorical features} categorical, {number of continuous features} numerical).~~ Your response should only contain the predicted accuracy in the format ## performance ##.
Hyperparameter configuration: {configuration 1}
Performance: {performance 1}
...
Hyperparameter configuration: {configuration n}
Performance: {performance n}
Hyperparameter configuration: {configuration to predict performance}
Performance:

---

Figure 16: LLAMBO [No context]. Prompt for the discriminative surrogate model with parts of the optimization problem description removed. Note that in this setting, no meta-data about the underlying datasets are provided. ~~Strikethrough~~ indicates the component that is removed.

---

The following are examples of the performance of a {model} measured in {metric} and the corresponding model hyperparameter configurations. ~~The model is evaluated on a tabular {task} task containing {number of classes} classes. The tabular dataset contains {number of samples} samples and {number of features} features ({number of categorical features} categorical, {number of continuous features} numerical).~~ The allowable ranges for the hyperparameters are: {configuration and type}. Recommend a configuration that can achieve the target performance of {target score}. Do not recommend values at the minimum or maximum of allowable range, do not recommend rounded values. Recommend values with the highest possible precision, as requested by the allowed ranges. Your response must only contain the predicted configuration, in the format ## configuration ##.
Performance: {performance 1}
Hyperparameter configuration: {configuration 1}
...
Performance: {performance n}
Hyperparameter configuration: {configuration n}
Performance: {performance used to sample configuration}
Hyperparameter configuration:

---

Figure 17: LLAMBO [No context]. Prompt for the candidate point sampler with parts of the optimization problem description removed. Note that in this setting, no meta-data about the underlying datasets are provided. ~~Strikethrough~~ indicates the component that is removed.

The following are examples of the performance of a {model} measured in {metric} and the corresponding model hyperparameter configurations. The model is evaluated on a tabular {task} task containing {number of classes} classes. The tabular dataset contains {number of samples} samples and {number of features} features ({number of categorical features} categorical, {number of continuous features} numerical). The allowable ranges for the hyperparameters are: {configuration and type}. Recommend a configuration that can achieve the target performance of {target score}. ~~Do not recommend values at the minimum or maximum of allowable range, do not recommend rounded values. Recommend values with the highest possible precision, as requested by the allowed ranges.~~ Your response must only contain the predicted configuration, in the format ## configuration ##.
Performance: {performance 1}
Hyperparameter configuration: {configuration 1}
...
Performance: {performance n}
Hyperparameter configuration: {configuration n}
Performance: {target score}
Hyperparameter configuration:

Figure 18: `LLAMBO` [No instructions]. Prompt for the candidate point sampler with additional instructions removed. ~~Strikethrough~~ indicates the component that is removed.

**Empirical analysis.** We assessed the end-to-end performance of our ablation configurations on Bayesmark tasks containing RandomForest models. The outcomes are illustrated in Figure 19. Our findings reveal that the standard `LLAMBO` configuration outperforms other variants, underscoring the significance of each prompt component in enhancing overall performance. Additionally, we note that our ablated settings achieve competitive optimization performances when compared against the baselines. This is more notable on `LLAMBO` [No context], which demonstrated similar optimization behavior without any meta-data about the underlying task. On one hand, this suggests the important role of meta-data in guiding the optimization process. For instance, specific hyperparameters better suited for large-p-small-n datasets can mitigate overfitting, and datasets with a higher proportion of categorical features may benefit from deeper decision trees. On the other hand, the robust performance of `LLAMBO` [No context], despite the lack of metadata,

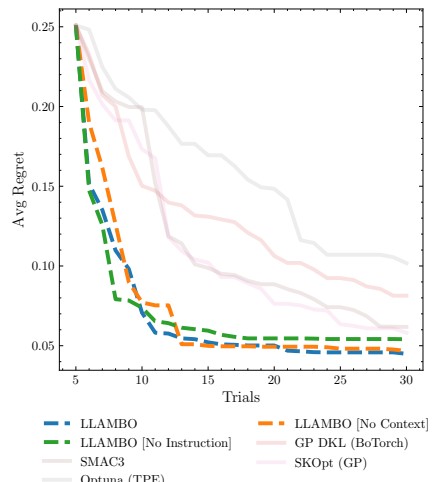

Figure 19: Ablation of prompt designs, averaged over 5 seeds.

signifies the model's effectiveness beyond mere reliance on data memorization or leakage, as it operates without access to specific dataset information.

Furthermore, our investigations into the role of candidate generation instructions indicate worse performance for `LLAMBO` [No instructions]. To understand this, we examined the acceptance rate of proposed points, defined as the proportion of points that both meet search space constraints and are unique (not duplicates of existing or other proposed points). `LLAMBO` [No instructions] recorded an acceptance rate of $69.26\% \pm 0.79\%$, significantly lower than `LLAMBO` ($91.60\% \pm 0.45\%$) and `LLAMBO` [No context] ($88.8\% \pm 0.39\%$). This reduced rate limits the effective pool of candidate points for evaluation and selection by the surrogate model. These findings underscore the importance of detailed task instructions in enhancing the quality and efficiency of the candidate sampling process, ultimately contributing to the overall optimization effectiveness.

## D  DETAIL OF EXPERIMENTAL PROCEDURES

In this section, we outline the benchmarks employed in our evaluations as well as the implementation details of our method and considered baselines. To evaluate the performance of LLAMBO on HPT,

we considered *25* built-in tasks from Bayesmark [31] (Appendix D.1, and *24* built-in tasks from HPOBench [32] (Appendix D.2). Additionally, we included three synthetic and three private datasets into Bayesmark, resulting in additional *30* tasks (Appendix D.3). We used *5* tasks extracted from Bayesmark (all 5 datasets on RandomForest) to evaluate each component, and used all tasks for end-to-end evaluations.

## D.1 BAYESMARK

We utilize Bayesmark [31] as a continuous HPT benchmark.[2] We included the 5 public datasets that came with the benchmark and 5 ML models, including RandomForest, SVM, DecisionTree, MLP, and AdaBoost. This makes a total of 25 tasks, with each task defined as the (dataset, model) pair. We execute all tasks using five different seeds for 25 trials, ensuring that all models share the same initialization for each seed. This approach guarantees consistency across results. For classification and regression tasks, the scoring function is accuracy and MSE, respectively.

**Hyperparameter space.** We follow the search space designated in Bayesmark, including the hyperparameter type, space, and range (lower and upper bound). We use this specified search space in all baselines, performing the transformations before the optimization process. The search space for each ML model is summarized below, i.e. {hyperparam_name: [{space}, {lower_bound}, {upper_bound}]}:

- SVM [$3d$]: {C: [log, 1, 1e3], $\gamma$: [log, 1e-4, 1e-3], tolerance: [log, 1e-5, 1e-1] }
- DecisionTree [$6d$]: {max_depth: [linear, 1, 15], min_samples_split: [logit, 0.01, 0.99], min_samples_leaf: [logit, 0.01, 0.49], min_weight_fraction_leaf: [logit, 0.01, 0.49], max_features: [logit, 0.01, 0.99], min_impurity_decrease: [linear, 0.0, 0.5] }
- RandomForest [$6d$]: {max_depth: [linear, 1, 15], min_samples_split: [logit, 0.01, 0.99], min_samples_leaf: [logit, 0.01, 0.49], min_weight_fraction_leaf: [logit, 0.01, 0.49], max_features: [logit, 0.01, 0.99], min_impurity_decrease: [linear, 0.0, 0.5] }
- MLP [$8d$]: {hidden_layer_sizes: [linear, 50, 200], alpha: [log, 1e-5, 1e1], batch_size: [linear, 10, 250], learning_rate_init: [log, 1e-5, 1e-1], power_t: [logit, 0.1, 0.9], tol: [log, 1e-5, 1e-1], momentum: [logit, 0.001, 0.999], validation_fraction: [logit, 0.1, 0.9]}
- AdaBoost [$2d$]: {n_estimators: [linear, 10, 100], learning_rate: [log, 1e-4, 1e1]}

## D.2 HPOBENCH

We included HPOBench, specifically the tabular benchmarks for computationally efficient evaluations [32]. We included all 8 OpenML datasets with pre-computed tabulated results, and 3 ML models, including XGBoost, RandomForest, and MLP. In total, we included 24 tasks. We execute all tasks using five different seeds for 25 trials, ensuring that all models share the same initialization for each seed. For all tasks, the scoring function is validation loss.

**Hyperparameter space.** The search spaces for all ML models, including their dimensionalities and search ranges, are provided below. The search spaces are discretized (see App D.3 in [32] for more details), allowing efficient tabular *look-up* operations for different configurations. For all baselines, the hyperparameters are treated as ordinal variables, following recommendations in the benchmark.

- XGBoost [$4d$]: {colsample_bytree: [0.1, 1.0], eta: [2e-10, 1], max_depth: [1, 50], reg_lambda: [2e-10, 2e10]}
- RandomForest [$4d$]: {max_depth: [1, 50], max_features: [0.0, 1.0], min_samples_leaf: [1, 2], min_samples_split: [2, 128]}
- MLP [$5d$]: {alpha: [1e-8, 1], batch_size: [4, 256], depth: [1, 3], learning_rate_init: [1e-5, 1], width: [16, 1024]}

## D.3 PRIVATE AND SYNTHETIC DATASET

Additionally, we introduce 3 proprietary (SEER [72], MAGGIC [73], and CUTRACT [74]) and 3 synthetic datasets. These are datasets for which the LLM would not have seen during pretraining,

---

[2]https://github.com/uber/bayesmark/tree/8c420e935718f0d6867153b781e58943ecaf2338

and thus used to address any memorization concerns. The synthetic datasets were generated from the complex multimodal functions, specifically: Rosenbrock, Griewank, and KTablet (see [75] for full simulation parameters). We selected the input dimension as 15, where each dimension is uniformly sampled in the range of $[0, 1]$. Subsequently, we used the designated functions in [75] to generate the corresponding output. All 6 datasets were introduced into Bayesmark, where the same set of ML models was evaluated (see Appendix D.1), leading to a total of 30 tasks. The other aspects of the evaluation were identical, with the same seeding, trials, scoring functions, and search spaces.

## D.4 IMPLEMENTATIONS

**End-to-end `LLAMBO`.** The end-to-end procedure iteratively performs three steps: **(1)** sample M candidate points $\{\tilde{h}_m\}_{m=1}^M$. **(2)** evaluate $M$ points using the surrogate model, i.e. $p(s|\tilde{h}_m)$ to obtain scores $\{a(\tilde{h}_m)\}_{m=1}^M$ according to an acquisition function. We use expected improvement (EI), $a(\tilde{h}_m) = \mathbb{E}[\max(p(s|\tilde{h}_m) - f(h_{best}), 0)]$. **(3)** select point with the highest score to evaluate next, $h = \arg\max_{\tilde{h} \in \{\tilde{h}_m\}_{m=1}^M} a(\tilde{h})$.

**Setting of `LLAMBO`.** For our instantiation of `LLAMBO`, we sample $M = 20$ candidate points, and set the exploration hyperparameter to $\alpha = -0.1$. $M = 20$ is similar to the default number of candidates sampled by popular TPE implementations, including HyperOpt, and Optuna.[34] We set the exploration hyperparameter to $\alpha = -0.1$ based on observations in Figure 5 as a value that balanced exploration (diversity) and exploitation. For the surrogate model, we sample $K = 10$ MC predictions to compute the empirical estimates. For our experiments, we used `gpt-3.5-turbo`, version `0301` with default hyperparameters `temperature` $= 0.7$ and `top_p` $= 0.95$.

**Baselines.** We select the following baselines:

- SKOpt (GP-based) [68]: We used the library implementation `https://scikit-optimize.github.io/stable/modules/generated/skopt.gp_minimize.html`. Additionally, we optimize the acquisition of candidates for better results, acq_optimizer = lbfgs. Version 0.9.0.
- GP (Deep Kernel Learning) [48]: We use the implementation of GPytorch `https://docs.gpytorch.ai/en/stable/examples/06_PyTorch_NN_Integration_DKL/KISSGP_Deep_Kernel_Regression_CUDA.html` in combination with the optimization of acquisition function used in BoTorch `https://botorch.org/docs/acquisition` for a better performance. (BoTorch version 0.8.5).
- DNGO: We used a public implementation `https://github.com/automl/pybnn`.
- SMAC3 [8]: We used version 1.4.0. from `https://github.com/automl/SMAC3`.
- Turbo [76]: We used the implementation found in `https://github.com/uber-research/TuRBO`.
- HEBO [77]: We used the available pip version of HEBO `https://pypi.org/project/HEBO/`. Version 0.3.5.
- Optuna [41]. We consider the implementation of `https://optuna.org/`. We used the multivariate version, which was found to have better performance in [44]. Version 3.3.0.
- TPE [75]. We incorporated an additional implementation of TPE, essentially a heavily optimized variant of TPE designed to achieve competitive results `https://github.com/nabenabe0928/tpe`.
- STO [39]. We used the official implementation provided by the authors `https://github.com/daizhongxiang/sto-bnts`.

# E ADDITIONAL RESULTS

## E.1 ADDITIONAL WARMSTARTING RESULTS

As part of our investigation into warmstarting, we investigated the effect of varying the amount of information provided in the zero-shot prompts on the quality of warmstart initialization. Specifi-

---

[3]`https://optuna.readthedocs.io/en/stable/reference/samplers/generated/optuna.samplers.TPESampler.html`

[4]`https://github.com/hyperopt/hyperopt/blob/master/hyperopt/tpe.py`

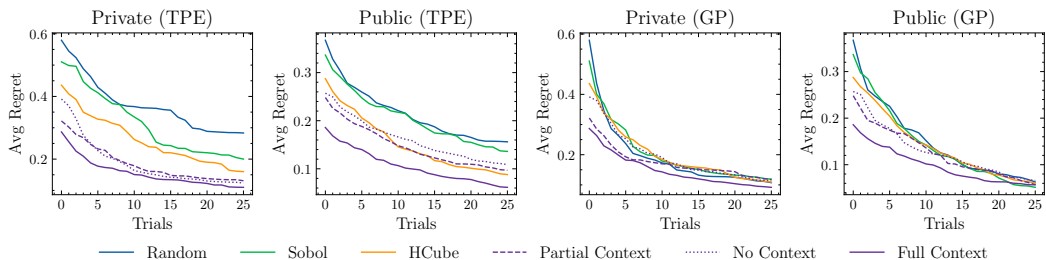

Figure 20: **Fine-grained warmstarting results.**

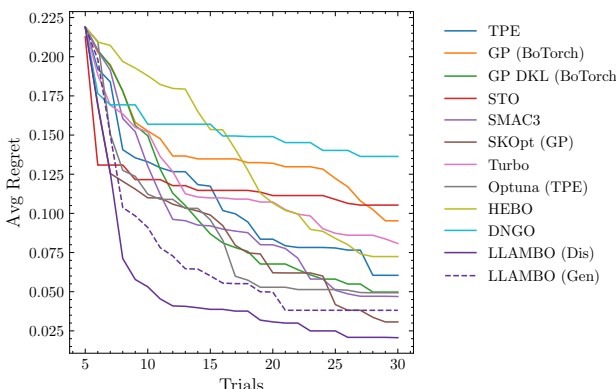

Figure 21: **End-to-end performance of `LLAMBO` (generative surrogate model) on Bayesmark.** Average regret on public, private, and synthetic datasets.

cally, ▶ **No Context**, ▶ **Partial Context** and ▶ **Full Context**. We evaluate the effectiveness of warmstarting by evaluating the search performance of GP and TPE under different initialization. In Figure 20, we plot average regret across tasks for both BO models. We observe similar trends, that increasing the informativeness of the prompts led to improved search performance. In Figure 25, we visualize the correlation matrices between sampled warmstarting points, observing higher correlations between points recommended by an LLM.

### E.2 LLAMBO WITH GENERATIVE SURROGATE MODEL

Due to budgetary constraints, we did not evaluate `LLAMBO` with both proposed surrogate models in Section 7. To determine the stronger surrogate model for our end-to-end evaluations, we tested both methods in a constrained setting. This meant assessing both models on a single run across all 25 tasks within the public Bayesmark benchmark. The outcomes of this preliminary evaluation are depicted in Figure 21. Our findings indicated that the `LLAMBO` with a discriminative surrogate model outperformed its counterpart. Consequently, we opted for the discriminative surrogate model for a comprehensive evaluation. However, it's worth noting that the `LLAMBO` using a generative surrogate model showcased competitive results when compared with baseline measures. Both approaches exhibited swift search and convergence in the initial trials, especially when $n < 10$. The generative instantiation was a close second until the last few trials.

Drawing definitive conclusions from a single seed can be challenging. However, we postulate that our model's generative version might exhibit sensitivity to the $\tau$ hyperparameter, which dictates the boundary between good and bad points. A recent study on inherent LLM biases in ICL by [64] underscored the majority label bias—a tendency of the model to favor the majority label in the given few-shot examples. This bias is intrinsically tied to our generative surrogate model's design, where $\tau$ determines the proportion of points labeled as good versus bad. We see a thorough examination of this potential bias, an in-depth analysis of the $\tau$ hyperparameter's sensitivity, and the exploration of bias-correction techniques as key future research avenues to fully explore the generative surrogate model's capabilities.

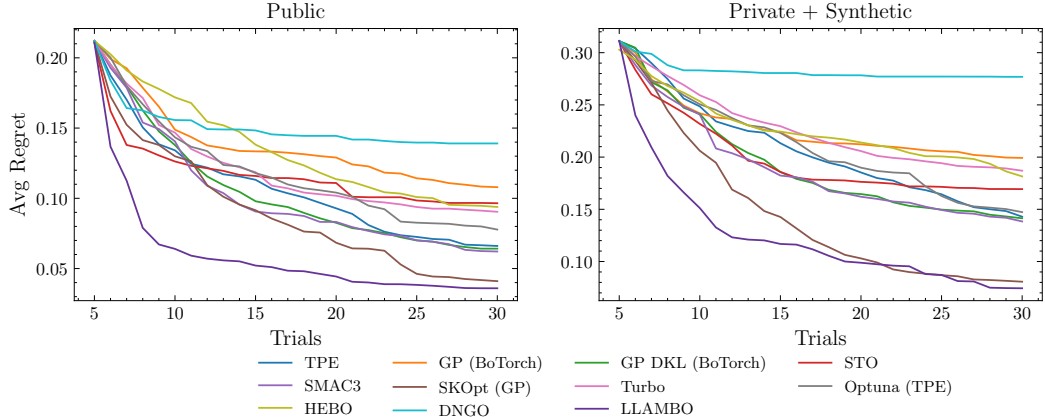

Figure 22: **End-to-end performance of `LLAMBO` on Bayesmark. (Left)** Average regret on public datasets and **(Right)** private and synthetic datasets aggregated on all tasks.

Table 1: Average rank (↓) achieved for different ML methods on public datasets.

|                  | DT   | MLP  | RF   | SVM  | ADA  |
|------------------|------|------|------|------|------|
| TPE              | 6.16 | 5.12 | 5.34 | **4.64** | 5.52 |
| GP (BoTorch)     | 6.72 | 5.62 | 5.94 | 6.06 | 4.74 |
| GP DKL (BoTorch) | 4.30 | 4.63 | 5.52 | 5.60 | 6.26 |
| SMAC3            | 5.82 | 6.42 | 5.40 | 5.66 | 6.02 |
| SKOpt (GP)       | 3.66 | **4.40** | 4.90 | 4.84 | 5.12 |
| Turbo            | 5.86 | **4.40** | 5.60 | 5.38 | 5.78 |
| Optuna (TPE)     | 6.58 | 5.34 | 5.48 | 5.08 | 5.08 |
| HEBO             | 6.14 | 4.88 | 5.10 | 6.30 | 5.72 |
| DNGO             | 7.74 | 8.62 | 8.08 | 6.62 | 7.14 |
| **`LLAMBO`**     | **2.02** | 5.06 | **3.64** | 4.82 | 3.62 |

Table 2: Average rank (↓) achieved for different ML methods on private and synthetic datasets.

|                  | DT   | MLP  | RF   | SVM  | ADA  |
|------------------|------|------|------|------|------|
| TPE              | 4.85 | 4.83 | 4.95 | 4.83 | **3.98** |
| GP (BoTorch)     | 6.70 | 6.18 | 7.08 | 7.10 | 5.45 |
| GP DKL (BoTorch) | 4.78 | 4.25 | 4.60 | 5.05 | 5.60 |
| SMAC3            | 5.12 | 5.33 | 5.13 | 6.15 | 5.40 |
| SKOpt (GP)       | 4.83 | 4.98 | 4.52 | **2.32** | 4.95 |
| Turbo            | 5.80 | 5.07 | 5.77 | 6.10 | 4.88 |
| Optuna (TPE)     | 4.98 | 5.75 | 5.37 | 5.12 | 5.52 |
| HEBO             | 6.50 | 6.33 | 6.25 | 6.13 | 5.15 |
| DNGO             | 7.83 | 8.17 | 9.12 | 7.32 | 7.62 |
| **`LLAMBO`**     | **3.35** | **4.10** | **2.08** | 4.88 | 6.45 |

### E.3 ADDITIONAL RESULTS ON BAYESMARK

In this section, we include additional results to supplement our end-to-end evaluation of Bayesmark.

**Additional baselines.** We include additional baselines in Figure 22, including an optimized version of TPE from [75], DNGO (a Bayesian neural network approach [24]), STO (a neural network approach [39]), Turbo [76] (a GP approach that identifies 'trusted regions' of the input space to search for improvements), and HEBO [77]. This presents a more comprehensive evaluation against a wide array of BO approaches.

**Search performance across ML models.** In Tables 1 and 2 we compare the tuning performance of BO methods for different types of ML models. We show the average rank achieved by methods at the end of each search. Although `LLAMBO` consistently demonstrates the best overall performance, it exhibits model-dependent variability. Notably, `LLAMBO` excels in tuning DecisionTree and RandomForest across both public and private benchmarks. However, it does less well on SVMs. The underlying cause of this remains speculative, but one plausible explanation is the inherent characteristics of the black-box function being optimized. This sensitivity and such nuances are common to all BO techniques, given their sensitivities to black-box functions with different attributes (e.g. the choice of kernel directly affects the effectiveness of GP for specific functions) [78]. Exploring the characteristics of black-box functions where our method shines is a crucial avenue for future research and remains beyond the scope of our current study.

**Individual task results.** We plot individual task results by optimization metric (Figure 26), regret (Figure 27), and average rank of BO methods (Figure 28) (all results are averaged over 5 runs).

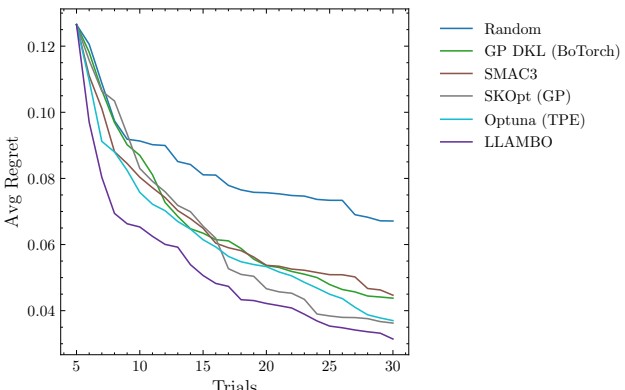

Figure 23: **End-to-end performance of `LLAMBO` (discriminative surrogate model) on HPOBench.** Average regret on all tasks across five seeds.

### E.4 Additional Results on HPOBench

In this section, we include additional results on HPOBench to supplement our end-to-end evaluation. This included a total of 24 tasks (where each *task* is a dataset-model pair, see Appendix D.2 for details). For each task, we executed 5 seeded search. Figure 23 shows the average regret across all tasks, we note that `LLAMBO` achieves the best tuning performance. Additionally, we observe that, consistent with prior findings, `LLAMBO` excels in earlier stages of the search, when fewer observations are available.

**Individual task results.** We plot individual task results by optimization metric (Figure 29), regret (Figure 30), and rank of `LLAMBO` and other BO methods (Figure 31).

### E.5 Clock Time

We analyze the runtimes of various algorithms. Figure 24 illustrates the average clock time per iteration as a function of the number of observed points. It is important to note that the times recorded represent solely the surrogate model's computation duration, excluding any black-box query time. For context, all runtime measurements were conducted on an Intel i7-1260P (a consumer-grade laptop). Our observations reveal that `LLAMBO` incurs a higher average clock time per iteration. However, it's essential to highlight that this increased time is predominantly influenced by external factors such as internet connectivity and API latency. Additionally, in black-box optimization scenarios, querying the black-box function is typically the primary computational expense, overshadowing the time required for Bayesian Optimization (BO). Therefore, optimizing the search process to minimize the number of iterations performed, and consequently, the number of black-box queries (i.e. sample efficiency), is arguably more significant than the time taken in each iteration of a BO algorithm.

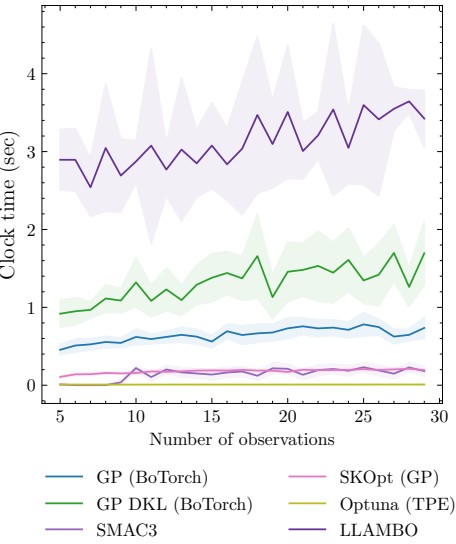

Figure 24: Average clock time (s) of BO algorithms.

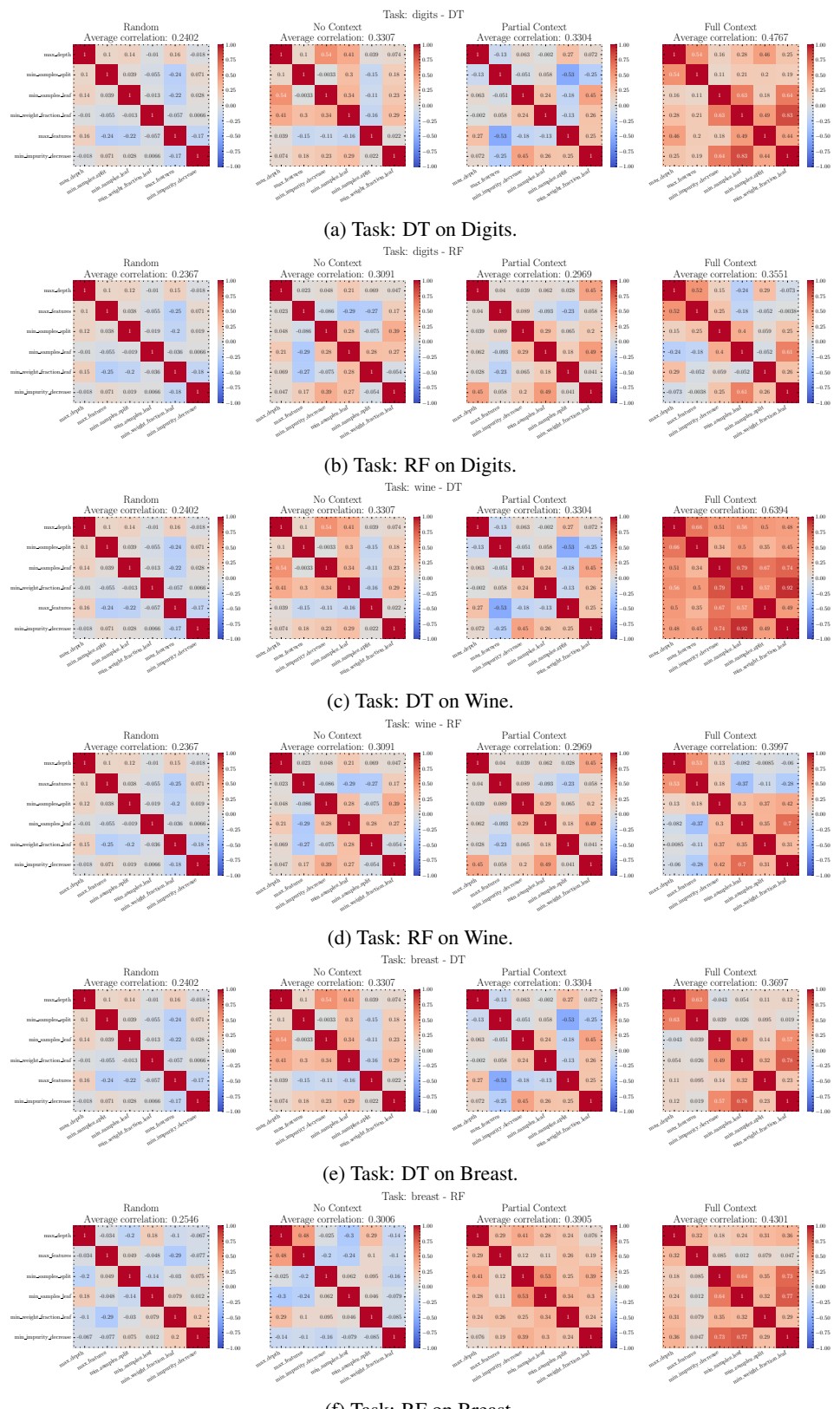

Figure 25: **Comparison of correlations between sampled initializations.** Correlation matrix calculated on 50 initialization sampled for each task.

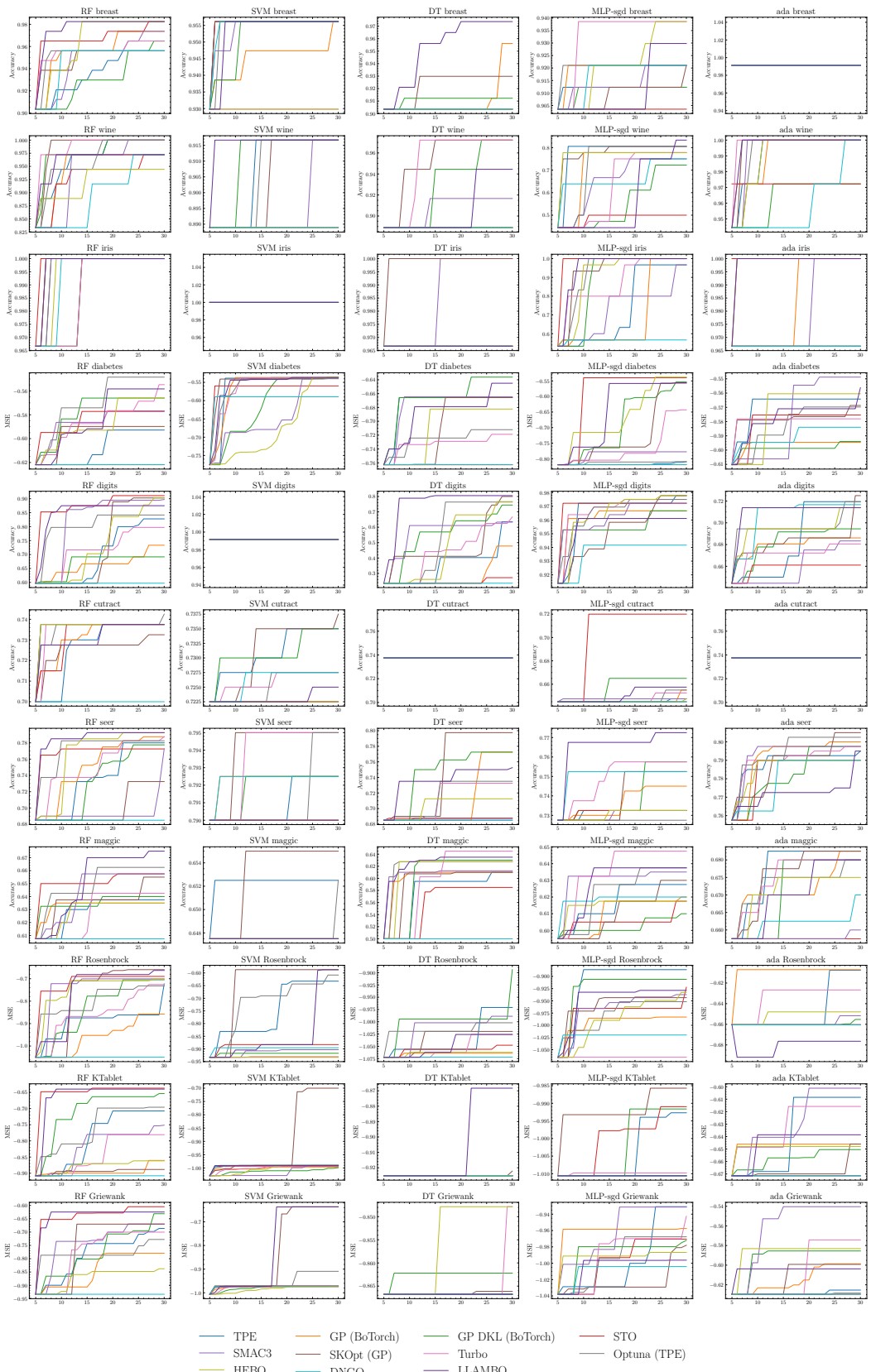

Figure 26: **Bayesmark: individual HPT task results (task metric).** Evaluations according to task metrics, i.e. accuracy (↑) for classification tasks, and *negative* MSE (↑) for regression tasks, averaged over 5 runs.

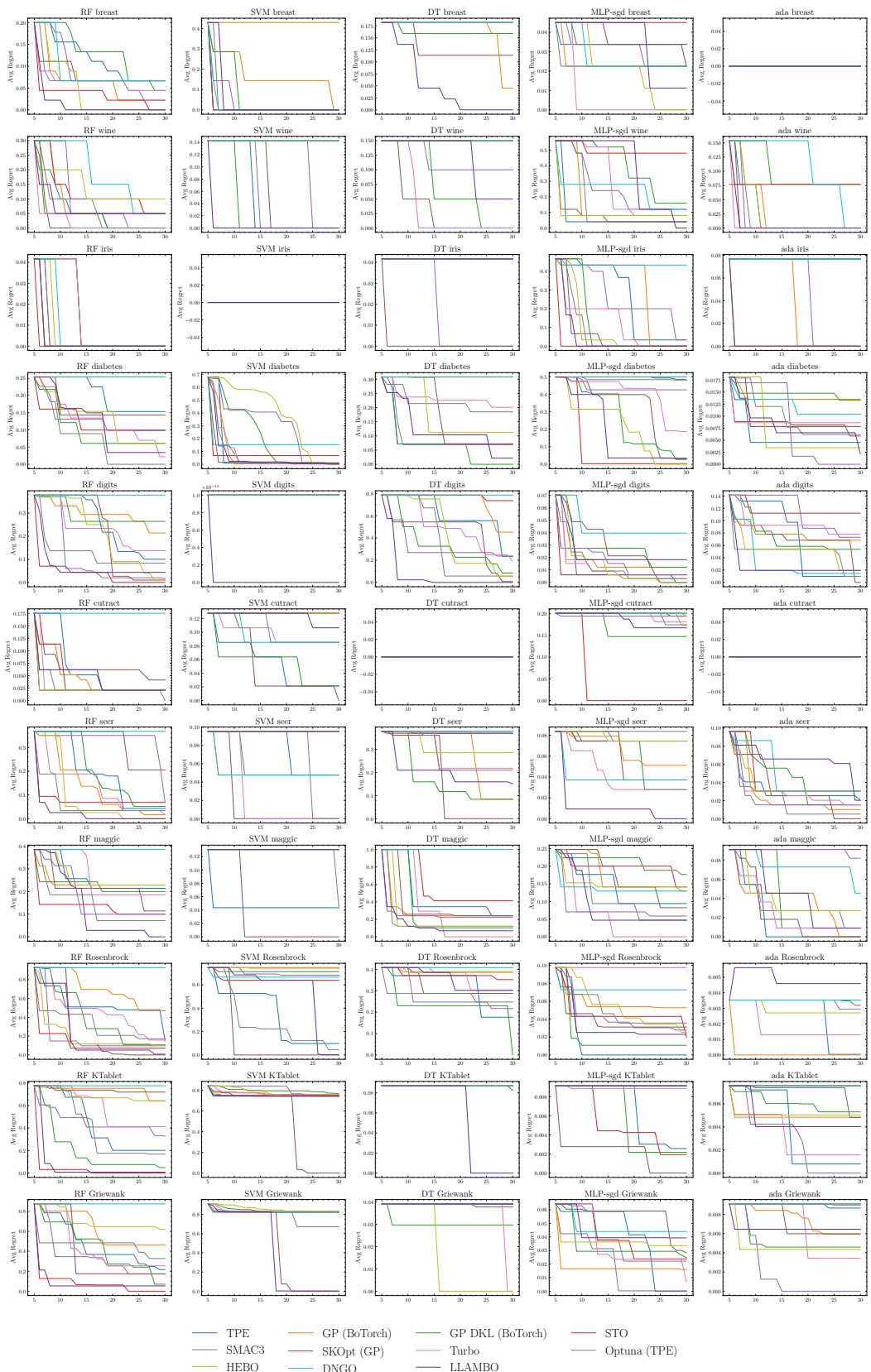

Figure 27: **Bayesmark: individual HPT task results (regret).** Evaluations according to normalized regret (↓), averaged over 5 runs.

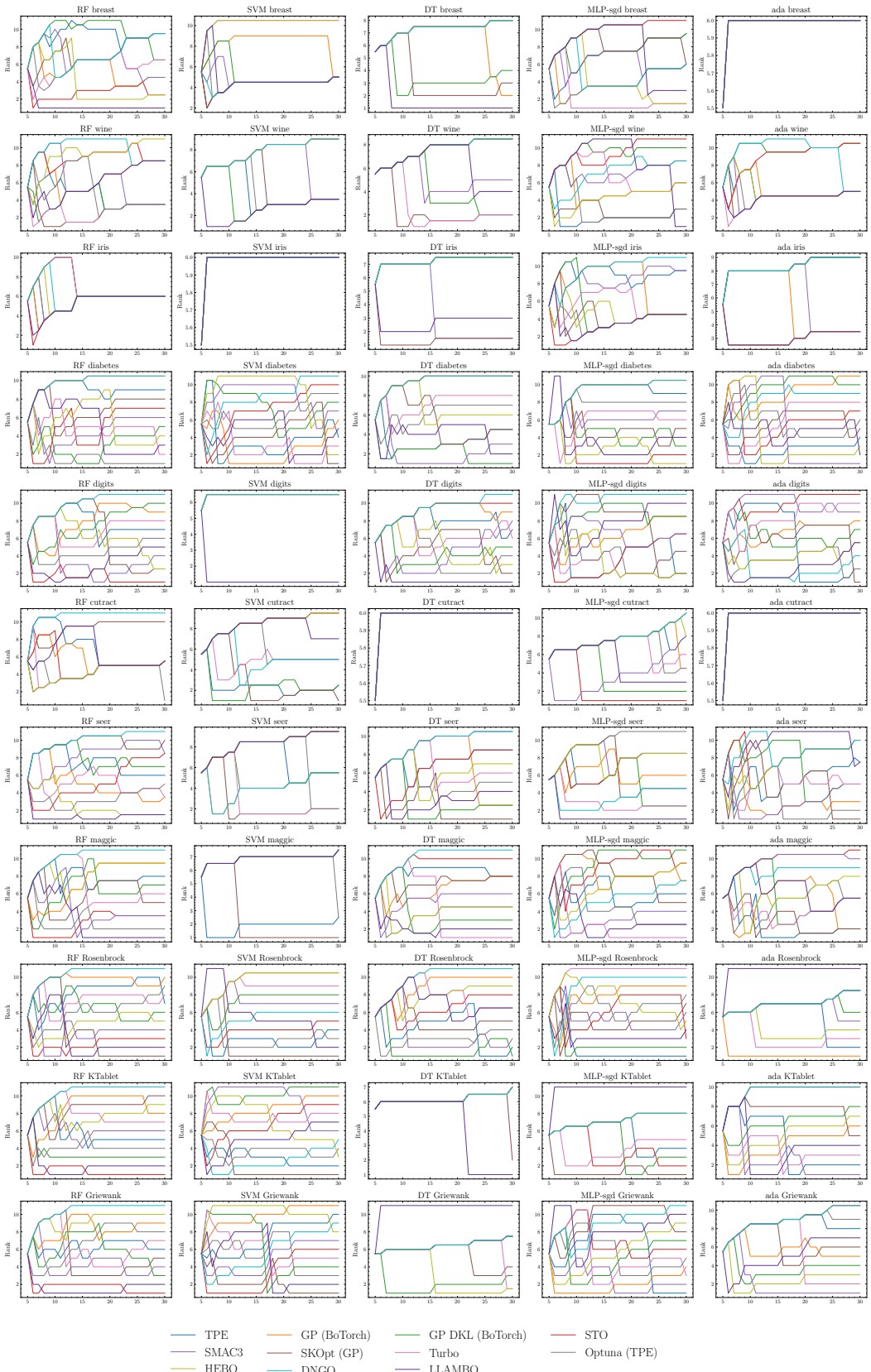

Figure 28: **Bayesmark: individual HPT task results (rank).** Evaluations according to average rank (↓) during each search, averaged over 5 runs.

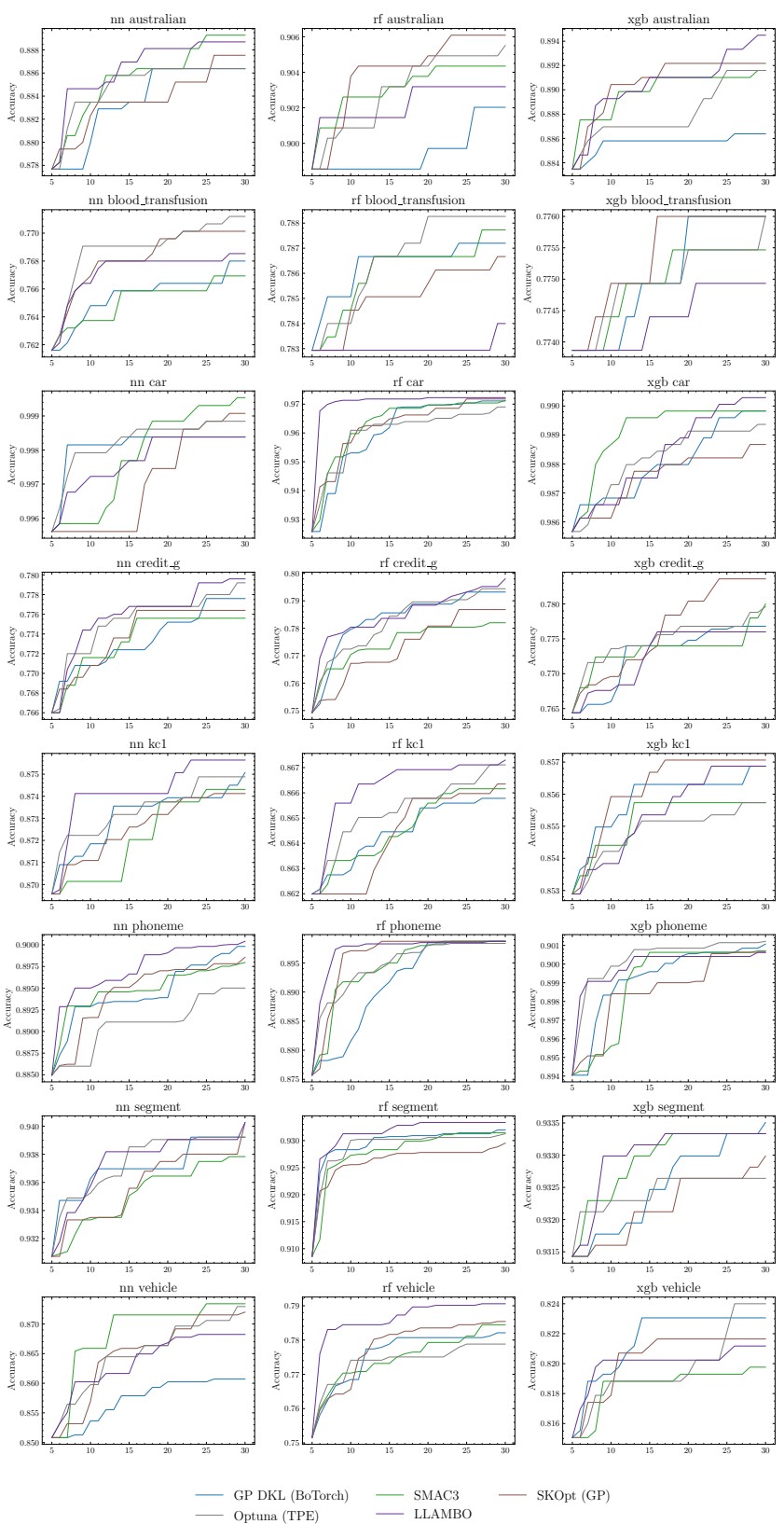

Figure 29: **HPOBench: individual HPT task results (task metric).** Evaluations according to task metrics, i.e. accuracy (↑), averaged over 5 runs.

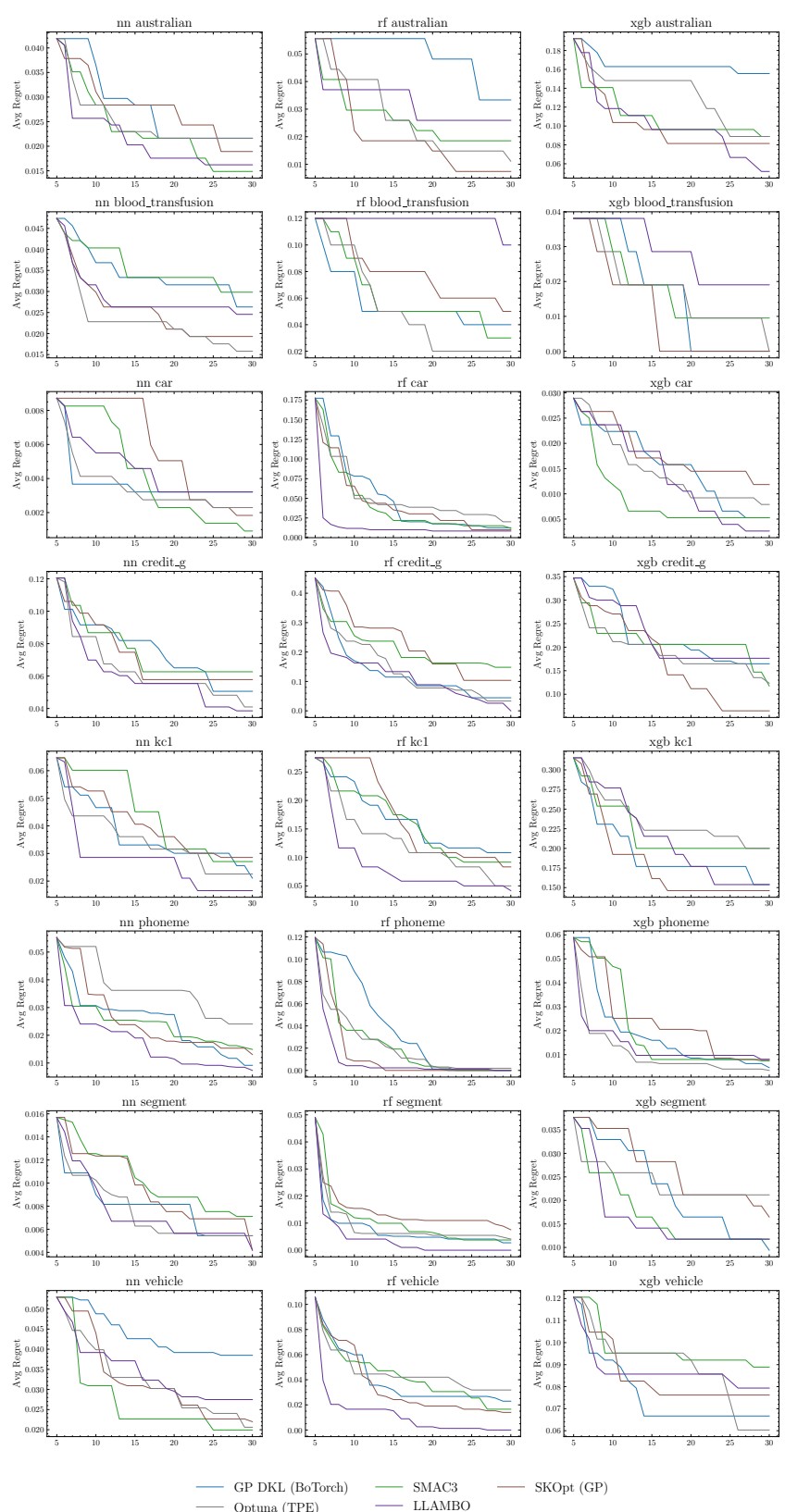

Figure 30: **HPOBench: individual HPT task results (regret).** Evaluations according to normalized regret (↓), averaged over 5 runs.

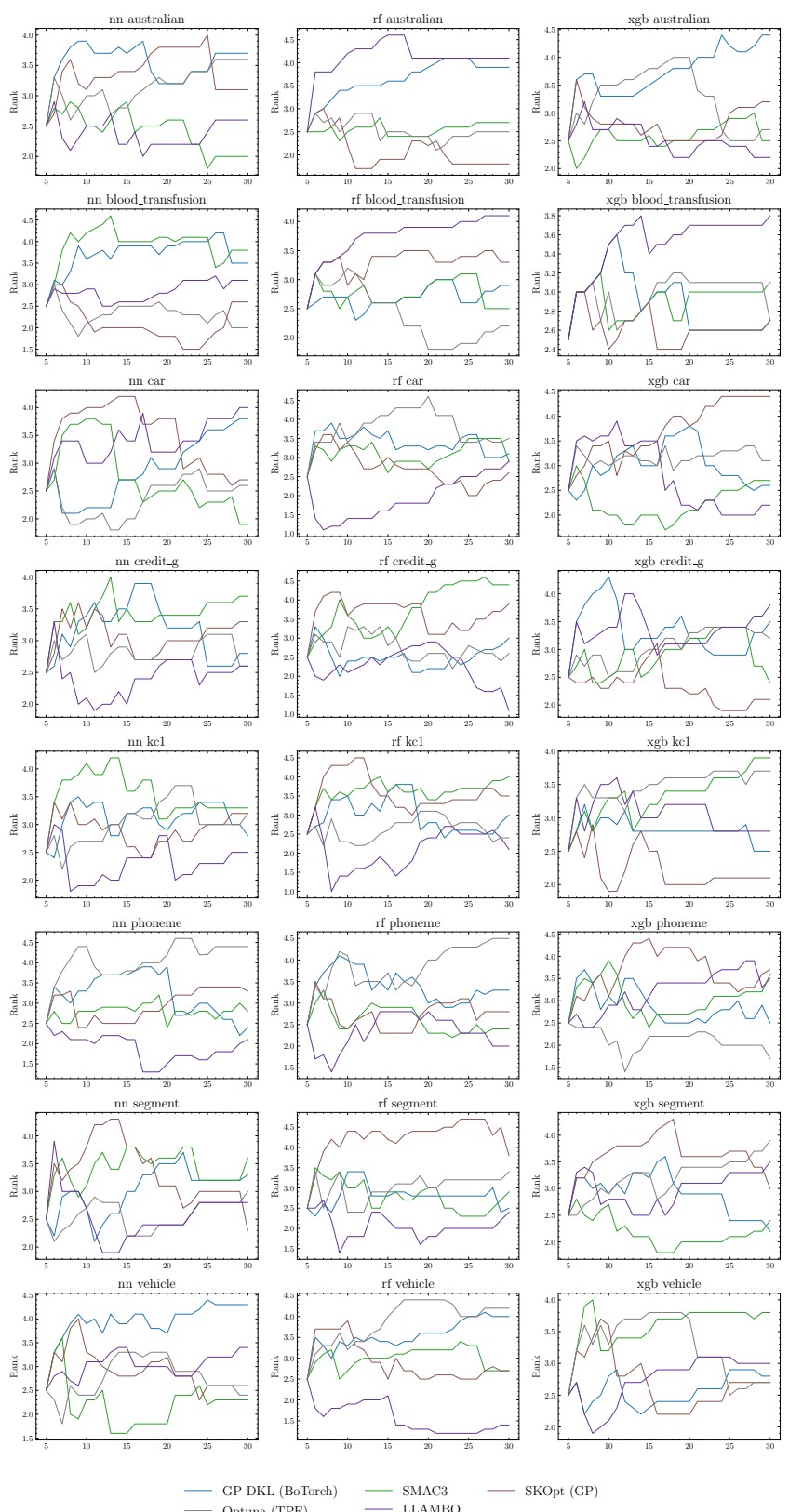

Figure 31: **HPOBench: individual HPT task results (rank).** Evaluations according to average rank (↓), averaged over 5 runs.

