# OpenReview forum: "Large Language Models to Enhance Bayesian Optimization"
_ICLR.cc/2024/Conference — ICLR 2024 poster_

### Official Review · Reviewer_8Jea · 2023-10-28

**Soundness:** 3 good
**Presentation:** 4 excellent
**Contribution:** 4 excellent
**Rating:** 8
**Confidence:** 5

**Summary:**

The authors present the new approach LLAMBO, which integrates large language models into Bayesian optimization for the case of hyperparameter optimization. The integration is done by translating knowledge about the problem, algorithm, and optimization history into natural language prompts. The modular approach comprises a warmstarting component, a candidate sampler, and a surrogate model. Two alternatives for the surrogate model are contrasted: a generative and a discriminative variant. The authors claim strong empirical performance to be shown in experimental evaluations. Each component is evaluated separately, and an evaluation of using all components together for an end-to-end approach in comparison to existing methods is carried out additionally.

**Strengths:**

Novelty & Significance: To the best of my knowledge, the approach is the first of its kind to integrate LLMs into the process of HPO/BO in this all-encompassing manner and could be interesting for the community to build upon.

Quality: In fact, the authors discuss many very interesting approaches for how to combine LLMs and BO. Eventually, they show that by combining all these approaches, they get a very strong system overall. Although I have some concerns regarding some details, the wealth of ideas in this paper and the corresponding experiments are impressive.

In the limited evaluation, the approach shows very promising results.

Clarity: The paper is written up nicely and illustrated with many figures and plots, making it relatively accessible.

**Weaknesses:**

### Approach
My main criticism is that the authors used an LLM that models data as a sequence of tokens. However, when passing HPO data to an LLM, they face the problem that this data has no sequential structure (see Section 4.1). They get around by permutating the data and thus even derive some kind of uncertainty. However, this seems flawed to me. The authors mention the SMAC method which is built on a random forest. Although random forests are not great for deriving uncertainties, the bootstrapping of random forests is at least statistically motivated. I missed any good argument as to why non-sequential data should be fed into a sequential model and then applying a hack trying to fix it again.

Furthermore, it is well known that the strength of LLMs is based on good prompts. Therefore, I would actually expect some kind of insights and ablation studies on how to do the prompts for this problem. Reading the exemplary prompt templates, I strongly wonder whether the authors came up with these prompts in their very first trial.

### Clarity

The paper is very dense and includes many nice results. However, the main paper (without appendix) should still be self-contained. However, the authors decided to move the discussion of related work into the appendix. I oppose that decision and deem it very important to have a discussion of related work in the main body of the paper s.t. readers can very well understand how to situate the paper. As a concrete proposal, I recommend moving the generative surrogate model into the appendix since it does not perform better than the discriminative model anyway.

### Experiments

First of all, all papers using closed-sourced GPT models have an inherent flaw: We know nothing about how these models are trained exactly, e.g., training data. Therefore, we lack any scientific understanding of how to use them. The authors tried to get around this by using some private and artificial datasets, but how does this relate to the training distribution used for the GPT model? We cannot make any real claim regarding meta-data leakage.

Furthermore, since GPT-3.5 is not publically available and might even be updated from time to time, the chance of reproducibility of these results is more or less not given at all. I strongly recommend using (at least) publicly available LLMs such as Llama 2.
There are many further doubts I will raise as questions below. These might translate to direct weaknesses if not properly answered in the rebuttal.

Some answers could relate to the use of Bayesmark. This benchmark library includes many rather simple HPO benchmarks, i.e., low-dimensional, continuous spaces and traditional ML models. Given state of the art in HPO, I would not consider these reasonable benchmarks anymore. In fact, quite some of the results in this paper conflict with insights into comparing HPO tools (see questions). Therefore, I suspect that results might look very different, if the authors would have used more challenging benchmarks (e.g, from HPOBench).

There are several further points that diminish the strength of the presented experimental results and undermine soundness:

* All plots use iterations instead of wall-time on the x-axis, which does not allow the reader to draw any conclusions on the overhead incurred through querying the LLM, which might negate any benefits incurred by its inclusion depending on the benchmark.
* No uncertainty is shown in any plot, even though results are averaged over several runs/ seeds.
* Especially in the end-to-end demonstration, the number of iterations chosen is very small, and it would be interesting to see how the curves continue.
* No fully random baselines were provided (warmstarting and end-to-end).
* There are some results that seem contradictory to previously observed behaviors of approaches (see questions)
* Accessibility: Often, only colors indicate which curves belong to which approach, and colors like red and green are mixed.

### Minor Points
* P. 15 (Appendix): 2. Candidate Point Sampler H, is "]" intentional?
* In Appendix E, Figure 16 the description hints to two graphs but only one is shown.
* In Figure 8, all curves start at the same point in iteration 0. This is a bit strange in comparison to the warmstarting, where there are clearly different starts depending on the methods. While this is probably due to all methods using the same warmstarting point, it seems puzzling to not include this at the start of the graph or make it clear from the description.
* In the plots, when regret or performance over time (iterations) is shown, the functions should be step functions since there are no interpolations in-between possible.

**Questions:**

1. How were the Hyperparameters of LLAMBO selected?
1. How were the hyperparameters of the other optimizers selected?
1. Which versions of the libraries for the other optimizers were used?
1. Which exact version of GPT-3.5 was used?
1. In Figure 7, for three of the four plots, the random approach does not seem to do anything, which seems questionable. Could you elaborate on why this might be the case?
1. Why was the end-to-end variant of LLAMBO not evaluated with its own warmstarting component?
1. In Figure 3, SMAC's random forest surrogate seems to beat the GP-based surrogate model, which based on previously published results seems unlikely given that we talk about a small-dimensional, continuous space. Could you elaborate on why this might be the case?
1. In Figure 8, SMAC seems to do nothing for the first few iterations, which, especially in comparison to the previous evaluation of the surrogates (e.g. vs GP) seems strange, could you elaborate on why this might be the case?
1. The performance of HEBO looks surprisingly bad compared to SMAC based on previously published results, could you elaborate on why this might be the case?

---

> ### Author Response · Authors · 2023-11-16
> **Response to Reviewer 8Jea (1/3)**
>
> *Thank you for your helpful feedback and comments. We aim to address all the individual points in your review here, but please also see the revised manuscript for changes (highlighted in teal).*
>
> ---
>
> ### [P1] LLMs and non-sequential data / empirical uncertainty
>
> **Modelling non-sequential data.** We thank the reviewer for raising this point. Using an LLM for optimization of non-sequential data is central to our paper, and how to best model non-sequential data forms the basis of our investigations. LLMs, while being sequential models, possess extensive prior knowledge, exhibit strong few-shot learning capabilities, and flexibly incorporate contextual information. These are desirable properties for BO algorithms. Our approach effectively taps into these LLM capabilities to enhance BO, albeit necessitating data serialization.
>
> We note that in other research domains, LLMs have been evaluated for processing non-sequential data. For example, [1-3], which demonstrated LLMs’ effectiveness on tabular regression, classification, and generation, respectively (where the data under consideration lacks sequential structure). More pertinent to our study, [4] employs a Transformer to sequentially process optimization results for HPO. Our empirical analysis further substantiates our method's effectiveness: despite data serialization, we can obtain a strong surrogate model (notably in few-shot scenarios, as shown in S4), along with an efficient candidate point sampler (S5) and an overall effective system (S6).
>
> Ultimately, the serialization of non-sequential data within our approach effectively elicits prior knowledge about problem space correlations, and capitalizes on the few-shot learning and contextual understanding capabilities of LLMs, resulting in an improved, sample-efficient search method. This advantage is vital in most BO applications, and we believe motivates our methodological choice. While performing BO with a sequential model is unconventional, we believe the strong performance and potential improvements of this novel approach is a valuable contribution and would inspire future research in this field.
>
> **Empirical uncertainty.** Just to clarify, we permute the order of in-context examples, not the hyperparameters within each example. While MC sampling for uncertainty estimation may lack the principled grounding of GP/RF, it is straightforward to implement and widely accepted technique. For example, in deep ensemble literature [5], empirical variance is used for uncertainty estimation. Conceptually, if we consider the predictive distribution produced by an LLM as being induced by the weights of the LLM ($\theta_{LLM}$) and the few-shot prompt ($\theta_{prompt}$), i.e. $p(y| x; \theta_{LLM}, \theta_{prompt})$ (note that $(\theta_{prompt}, x)$ represents the complete prompt), our MC sampling is an approximation of the expectation $E_{\theta_{prompt}\sim p(\theta_{prompt})} [p(y|x; \theta_{LLM}, \theta_{prompt})]$, which marginalizes over the distribution of different prompts. Note, this is conceptually identical to the approach in deep ensembles, where $E_{w\sim p(w)}[p(y|x; w)]$. Intuitively, each differently prompted LLM can be considered an ensemble member, where we compute empirical variance from the ensemble. Indeed, we demonstrated that this approach of combining MC sampling with permutations of in-context examples improved calibration of uncertainty estimates, enhancing end-to-end performance.
>
> [1] Dinh, T., Zeng, Y., Zhang, R., Lin, Z., Gira, M., Rajput, S., Sohn, J.Y., Papailiopoulos, D. and Lee, K., 2022. Lift: Language-interfaced fine-tuning for non-language machine learning tasks. Advances in Neural Information Processing Systems, 35, pp.11763-11784.
>
> [2] Hegselmann, S., Buendia, A., Lang, H., Agrawal, M., Jiang, X. and Sontag, D., 2023, April. Tabllm: Few-shot classification of tabular data with large language models. In International Conference on Artificial Intelligence and Statistics (pp. 5549-5581). PMLR.
>
> [3] Borisov, V., Sessler, K., Leemann, T., Pawelczyk, M. and Kasneci, G., 2022, September. Language Models are Realistic Tabular Data Generators. In The Eleventh International Conference on Learning Representations.
>
> [4] Chen, Y., Song, X., Lee, C., Wang, Z., Zhang, R., Dohan, D., Kawakami, K., Kochanski, G., Doucet, A., Ranzato, M.A. and Perel, S., 2022. Towards learning universal hyperparameter optimizers with transformers. Advances in Neural Information Processing Systems, 35, pp.32053-32068.
>
> [5] Lakshminarayanan, B., Pritzel, A. and Blundell, C., 2017. Simple and scalable predictive uncertainty estimation using deep ensembles. Advances in neural information processing systems, 30.

---

> ### Author Response · Authors · 2023-11-16
> **Response to Reviewer 8Jea (2/3)**
>
> ### [P2] Prompt design
>
> Thank you for this suggestion. Our prompt design adheres to the framework outlined in S2.2, including three key components: (1) optimization problem description, (2) optimization history, and (3) explicit task instructions. We would like to re-emphasize that the primary objective of our research as to *investigate the feasibility* of enhancing BO with LLMs, and not designing highly robust or optimized prompts. Most of our efforts on prompt design were concentrated on (3), ensuring that LLAMBO behaves robustly from a system perspective, consistently generating predictions/samples in valid formats amenable for subsequent processing.
>
> To further shed light on this process, we have further elaborated on the prompt design process in App C.2. This also includes a new ablation study wherein we methodically evaluated the impact of various components, specifically **(a)** omitting meta-data from context description (c.f. (1)) and **(b)** specific task instructions (c.f. (3)). Our ablation revealed that standard LLAMBO configuration outperformed other variants, underscoring the significance of each prompt component in enhancing overall performance. Our findings on **(a)** indicated that meta-data played a hand in improving search performance further, but excluding it did not significantly affect performance. On examining **(b)**, we also found that the task-specific instructions improved validity of proposed candidate points, leading to more efficient search. We elaborate on **(a)** further in **[P4]** below.
>
> However, we view combining recent advances in automatic prompt tuning within our framework as a promising direction that could further improve the optimization performance of LLAMBO.
>
> **Actions taken.** We have improved our manuscript with a (1) description of the prompt design process and (2) prompt ablation study in App C.
>
> ---
>
> ### [P3] Clarity
>
> We appreciate this concrete recommendation, which we have followed by reorganizing the paper to include related works in the main text.
>
> ---
>
> ### [P4] Meta-data leakage
>
> We acknowledge your concern regarding the potential for meta-data leakage, particularly the risks of LLM memorizing task-specific hyperparameters. To mitigate this, our evaluation included three private and three synthetic datasets. It is crucial to note that no *semantic* information that could be used to directly identify underlying datasets is provided (please see our prompts in App C). Our synthetic datasets, while generated from known DGPs, had controllable input dimensions and sampling distributions, set by us.
>
> Regarding the meta-data in our prompts, we provided only high-level information, inclusively: [# of # of features (# categorical, # numerical), # of samples, and tasks described as either classification or regression]. This information is provided as HPT optimization should benefit from this information, for instance, to avoid overfitting on big-p-little-n tasks.
>
> To concretely address your concerns, we point to three pieces of empirical evidence:
> * In our warmstarting experiments (S3), we analyzed a no context zero-shot prompt, where no semantics or meta-data is provided (see App C), and observed the LLM was able to suggest configurations that exhibited stronger correlations (indicating dataset-agnostic knowledge of generalizable correlations in the optimization landscape) that improved optimization performance.
> * In S4.2 and S5, we performed ablations that removed the aforementioned meta-data from the prompts **LLAMBO (UnInf)** and found that, while the performance degraded relative to the full prompt, the performance was consistently superior to considered baselines (see Figure 4, 6).
> * **[New result]** To further address any concerns of meta-data leak, we perform an additional ablation study, where we removed any meta-data from the prompts and analyzed resulting end-to-end performance. This is included in App C.2. Expectedly, this ablation performs slightly worse, as meta-data can improve hyperparameter suggestion (e.g. specific hyperparameters better suited for large-p-small-n  to guard against overfitting). However, we note that the ablation was consistently superior against compared baselines, and achieved very similar optimization performance as the full setting. This highlights LLAMBO’s effectiveness despite the lack of metadata, indicating the performance goes beyond mere reliance on data memorization or leakage (as it operates without access to specific dataset information).
>
> We hope this sufficiently addresses your concerns about meta-data leakage.
>
> **Actions taken.** In response to your feedback, we have conducted an ablation study to assess the impact of including meta-data in prompts on end-to-end performance in App C.

---

> ### Author Response · Authors · 2023-11-16
> **Response to Reviewer 8Jea (3/3)**
>
> ### [P5] Reproducibility
>
> We are committed to ensuring full reproducibility of our results. To ensure this, we will release code, prompt templates, and search history/results for all 79 tasks upon acceptance of the paper. This will allow future research to benchmark against our findings without needing access to our exact API endpoints.
>
> While we acknowledge the reproducibility of the results will depend on the underlying LLM, we would like to highlight the generality of our approach, emphasizing that the novelty and contributions are not in the capabilities of any *specific LLM*, but in the notion of using LLM for model-based BO. Ultimately, as LLMs improve, we believe so will the performance of our proposed approach.
>
> ---
>
> ### [P6] Additional benchmarks
>
> We appreciate your recommendation. Accordingly, we have expanded our analysis to include 24 additional tasks (8 datasets, 3 ML models) from HPOBench in Appendix E.4. In the interest of time, we note that we have only included results from one seeded run, but we plan to include results across 5 runs (which will be updated as they come in). Our expanded analysis reveals LLAMBO achieves strong performance across this extended set of results, further validating the efficacy and generalizability of our approach.
>
> ---
>
> ### [P7] Additional comments about experiments/writing
>
> * **Clock time.** We have now included average clock time for LLAMBO and baselines in Appendix E.5.
> * **Uncertainty.** We have updated all figures to include 95% CI.
> * **Random baseline.** We have incorporated random search as a baseline in our end-to-end results in Figure 7.
> * **Typos.** These have been fixed.
> * **Figure 8.** Your interpretation is correct, all baselines receive the same initialization points (and we highlighted this in S6).
>
> ---
>
> ### [P8] Questions
>
> 1. We sample $M=20$ candidate points, generate $K=10$ MC predictions, and set the exploration hyperparameter to $\alpha=-0.1$. The choice of $M=20$ aligns with the default/recommended number of candidates sampled in popular TPE implementations (including HyperOpt, Optuna). We set $\alpha=-0.1$ based on observations described in Figure 6 as a value that balanced diversity-performance.
> 2. We utilized default meta-hyperparameters included in baseline optimizers.
> 3. These details have been included in Appendix D.
> 4. gpt-3.5-turbo version 0301.
> 5. The random baseline samples the same set of M=20 points for each seed, irrespective of the number of observed samples. This accounts for the constant regret and diversity across different values of N.
> 6. To ensure an equitable comparison with other optimizers, we did not employ warmstarting in our end-to-end evaluation. All methods, including ours, were initialized with the same set of points.
> 7. When evaluated solely as a surrogate model, SMAC exhibits better predictive performance at lower values of $N$. A possible explanation is overfitting in GP with smaller $N$, whereas RF will benefit from the variance reduction effect from ensembling. At higher values of $N$, however, GP demonstrates enhanced uncertainty estimates and predictions, leading to better overall performance in end-to-end evaluations.
> 8. We apologize for a plotting error in the SMAC curve that incorrectly included initialization points in the initial 5 iterations. This has now been corrected in average results in Figure 7, 20, 21, but also individual task results in Figures 25-27.
> 9. We used the pip available implementation of HEBO and default meta-hyperparameters. The original HEBO paper reports results from either batch mode (8 points per iteration) for 16 iterations or (1 point per iteration) for 100 iterations, which differs from the experimental settings in our work. Additionally, we noted similar findings that while HEBO was highly competitive given enough iterations, it has been observed to be inefficient in earlier iterations (e.g. Figure 4 of [1]).
>
> [1] Müller, S., Feurer, M., Hollmann, N. &amp; Hutter, F.. (2023). PFNs4BO: In-Context Learning for Bayesian Optimization. <i>Proceedings of the 40th International Conference on Machine Learning</i>, in <i>Proceedings of Machine Learning Research</i> 202:25444-25470 Available from https://proceedings.mlr.press/v202/muller23a.html.
>
> ---
>
> We thank the reviewer for your help in improving our work. Please let us know if our latest changes have addressed your concerns, and if there is anything else you would like to see.

---

> > ### Comment · Reviewer_8Jea · 2023-11-16
> > **Thank you for the reply**
> >
> > Dear Authors,
> >
> > Thank you very much for your thorough and impressive reply and revision. Many of my concerns are solved by this.
> >
> > Nevertheless, I'm not entirely convinced by the following argument:
> >
> > > emphasizing that the novelty and contributions are not in the capabilities of any **specific** LLM,
> >
> > How do you know that? Maybe some other LLMs will perform much worse or even better.
> > Why don't you run e.g. LLAMA2 in addition? This would also strengthen the reproducibility quite a bit.
> >
> > I'm also not very much in favor of the new sentence in the revision:
> >
> > > Our selection of this model is based on its cost-effectiveness, ease of querying, and performance capabilities
> >
> > This sounds more like a marketing argument and less like scientific reasons. If something like performance capabilities were your main concern, I would have expected you to go for GPT-4 and not 3.5.

---

> > > ### Author Response · Authors · 2023-11-16
> > > **Thank you and follow up response**
> > >
> > > *Thank you for your swift response. We are pleased to have addressed many of your concerns.*
> > >
> > > The core message of this work is to showcase the potential of LLMs in BO, and to provide insights into their effective integration within the BO process. We wish to clarify that our claim is not that LLAMBO will work with *every* LLM, but rather that LLMs possess the potential for successful BO application.
> > >
> > > In this context, we believe that demonstrating our approach’s effectiveness with *an* LLM (in this case, GPT-3.5), is sufficient. Our aim is not to compare various LLMs in BO, as we agree that performances will inevitably vary—some may perform worse, others better. This aspect, although outside our current scope, presents an exciting avenue for future research. Particularly, as LLMs gain traction in BO and other optimization tasks, understanding the strengths and weaknesses of different models becomes crucial. Identifying the most suitable LLM for specific scenarios, or even combining multiple LLMs to enhance optimization performance, are exciting possibilities. Future studies could benefit from benchmarks similar to those that have been proposed for [question answer (HellaSwag)](https://arxiv.org/abs/1905.07830) and [mathematical reasoning (GSM8K)](https://arxiv.org/abs/2110.14168).
> > >
> > > Lastly, we appreciate your input on the new sentence and agree to exclude it from the revised manuscript as you recommended. We have incorporated the aforementioned discussion regarding the choice of LLMs into the Discussion section of our paper.
> > >
> > > With thanks,
> > >
> > > Authors of #8831

---

> > > > ### Comment · Reviewer_8Jea · 2023-11-16
> > > > **Changing the Rating**
> > > >
> > > > Thanks. I see your point, but nevertheless, I believe any open-source LLM is a much better decision than GPT-3.5. Nevertheless, I agree with your overall arguments and I see sufficient room for further studies on the influence of different LLMs. Therefore, I increased my score accordingly.

---

> > > > > ### Author Response · Authors · 2023-11-16
> > > > > **Thank you**
> > > > >
> > > > > We are delighted we were able to address your concerns. We appreciate your constructive feedback and concrete guidance, which enabled us to improve our paper’s scope, contributions, and empirical analyses, enhancing the quality of our work.

---

### Official Review · Reviewer_LVtK · 2023-10-31

**Soundness:** 3 good
**Presentation:** 3 good
**Contribution:** 3 good
**Rating:** 8
**Confidence:** 3

**Summary:**

This paper devises a method to enhance Bayesian optimization using a large language model.  By employing GPT-3.5, the authors investigate GPT-3.5's abilities to warm-start Bayesian optimization, model a surrogate function, and sample query points, and eventually conduct the entire process of Bayesian optimization.  In this paper various scenarios are conducted using in-context learning and prompt engineering, and some important messages discovered by the authors are delivered.

**Strengths:**

* It solves an interesting topic in Bayesian optimization or active search.  Many optimization researchers were curious about the topic handled in this work.
* Many questions on how it works and which factor makes it work are answered and discussed.
* Extensive analyses are provided.
* Paper is generally well-written.

**Weaknesses:**

* I think this is timely work, but I am not sure that it can be presented at ICLR, which is a conference that focuses on representation *learning*.  This work did not do learning explicitly.  I am leaning towards a positive side, but it should be carefully discussed with authors, reviewers, and area chairs.  I think that this paper is more suitable for a natural language processing conference such as ACL, EMNLP, or others.
* Standard deviation (or confidence interval) is not reported for every experiment.  I think this is an important component for this kind of studies.  If some results are statistically meaningless, the analyses are not much meaningful.
* I am curious about how the authors design prompt templates.  Since I tried similar experiments, I could understand why some sentences are included.  However, the analysis on prompts and prompt designs, and potentially failure cases, should be included in the paper.  For example, if you do not include "Do not recommend values at the minimum or maximum of allowable range, do not recommend rounded values" in the prompt, what happens?

**Questions:**

* In prompt examples, do colors have some consistent indication?  For example, texts in orange have some meaning?  I think you can make consistency across examples and it can help understand the prompt examples.
* Why is no context for warm-starting Bayesian optimization better than random, sobol, or hcube?  The results of no context should be similar to the random initialization methods.
* In discriminative surrogate modeling, a method of LLAMBO utilizes both Monte Carlo sampling and shuffling of examples, or it only uses the shuffling?
* Could you elaborate how $\alpha$ is used in candidate point sampling?
* Could you explain the details of end-to-end demonstration of LLAMBO?  I think it is missing in the paper including the appendices.
* As I mentioned earlier, I would like to see the design process of prompts.

---

> ### Author Response · Authors · 2023-11-16
> **Response to Reviewer LVtK (1/2)**
>
> *Thank you for your helpful feedback and comments. We aim to address all the individual points in your review here, but please also see the revised manuscript for changes (highlighted in teal).*
>
> ---
>
> ### [P1] Relevance to ICLR
>
> Thank you for raising your concern. We believe that our work aligns well with the thematic areas outlined in the [CfP](https://iclr.cc/Conferences/2024/CallForPapers), particularly with the domains of BO, LLM prompting, and more generally transfer learning and applications of LLMs. We consider our work as a contribution to BO, a key enabler in many ML fields and core topic in ICLR, as evidenced by prior publications [1-3]. Additionally, research on application and prompting of LLMs are also commonly featured at ICLR [4-6]. We believe that our findings not only demonstrate the feasibility of integrating LLMs to enhance BO, but also open up avenues for future research in enhancing optimization algorithms more generally, making it highly pertinent to the themes and community at ICLR.
>
>
> [1] Wistuba, M. and Grabocka, J., 2020, October. Few-Shot Bayesian Optimization with Deep Kernel Surrogates. In International Conference on Learning Representations.
>
> [2] Sussex, S., Makarova, A. and Krause, A., 2022, September. Model-based Causal Bayesian Optimization. In The Eleventh International Conference on Learning Representations.
>
> [3] Volpp, M., Fröhlich, L.P., Fischer, K., Doerr, A., Falkner, S., Hutter, F. and Daniel, C., 2019, September. Meta-Learning Acquisition Functions for Transfer Learning in Bayesian Optimization. In International Conference on Learning Representations.
>
> [4] Zhou, D., Schärli, N., Hou, L., Wei, J., Scales, N., Wang, X., Schuurmans, D., Cui, C., Bousquet, O., Le, Q.V. and Chi, E.H., 2022, September. Least-to-Most Prompting Enables Complex Reasoning in Large Language Models. In The Eleventh International Conference on Learning Representations.
>
> [5] Zhou, Y., Muresanu, A.I., Han, Z., Paster, K., Pitis, S., Chan, H. and Ba, J., 2022, September. Large Language Models are Human-Level Prompt Engineers. In The Eleventh International Conference on Learning Representations.
>
> [6] Yao, S., Zhao, J., Yu, D., Du, N., Shafran, I., Narasimhan, K.R. and Cao, Y., 2022, September. ReAct: Synergizing Reasoning and Acting in Language Models. In The Eleventh International Conference on Learning Representations.
>
> ---
>
> ### [P2] Reporting uncertainty
>
> Thank you for this suggestion. We originally omitted error bars to improve visual clarity in interest of limited space. We have since revised the manuscript to include 95% CI on all plots  (Fig 2 - Fig 8).
>
> ---
>
> ### [P3] Prompt design
>
> Thank you for this suggestion. Our prompt design adheres to the framework outlined in S2.2, including three key components: (1) optimization problem description, (2) optimization history, and (3) explicit task instructions. Given that the primary objective of our research was to *investigate the feasibility* of enhancing BO with LLMs, we did not prioritize extensive refinement of the prompts design. Most of our efforts on prompt design were concentrated on (3), ensuring that LLAMBO behaves robustly from a system perspective, consistently generating predictions/samples in valid formats amenable for subsequent processing.
>
> To further shed light on this process, we have further elaborated on the prompt design process in App C.2. This also includes a new ablation study wherein we methodically evaluated the impact of various components, specifically **(a)** omitting meta-data from context description (c.f. (1)) and **(b)** specific task instructions (c.f. (3)). Our ablation revealed that standard LLAMBO configuration outperformed other variants, underscoring the significance of each prompt component in enhancing overall performance. Our findings on **(a)** indicated that meta-data played a hand in improving search performance further, but excluding it did not significantly affect performance. On examining **(b)**, we also found that the task-specific instructions improved validity of proposed candidate points, leading to more efficient search.
>
> Specifically to respond to your question: *“do not recommend values at the minimum or maximum of allowable range, do not recommend rounded values”*, these are part of (3). We included these instructions to prevent the LLM from suggesting candidate points that were under-precise or snapped to the limits of the search space. We found that this instruction significantly improved the percentage of suggested points that were valid (i.e. conform to search space) and unique (i.e. not duplicated). Please refer to C2 for a more in-depth discussion.
>
> **Actions taken.** We have improved our manuscript with a (1) description of the prompt design process and (2) prompt ablation study in App C.

---

> ### Author Response · Authors · 2023-11-16
> **Response to Reviewer LVtK (2/2)**
>
> ### [P4] Questions
>
> * **Color legend.** Thank you for this suggestion, and have incorporated a color legend in the revised manuscript.
> * **No context warmstarting.** No context warmstarting is expected to be better than random initialization. In fact, this was one of our initial sanity check to verify that LLMs contained useful prior knowledge about the optimization problem. This strategy relies on LLMs' intrinsic knowledge of hyperparameter correlations that are independent of specific problems. For instance, there are generalizable correlations, such as the interplay between tree depth and the number of estimators in Random Forests, which are crucial for managing the bias-variance tradeoff. Unlike random initialization methods, no context warmstarting exploits these inherent correlations identified by LLMs, thereby enhancing the optimization process.
> * **Discriminative SM.** LLAMBO incorporates both MC sampling and permutation of in-context examples. This is in contrast to the **LLAMBO (MC)** variant, which exclusively relied on MC sampling. Our comparative analyses demonstrated that permutations improved calibration of uncertainty estimates obtained from MC sampling. We have updated the manuscript to be more clear about this distinction.
> * **$\alpha$.** The candidate sampler generates points, denoted as $h$, from the distribution $p(h|s’, \mathcal{D}\_n)$. This process is conditioned on a desired target value $s’$ and the existing observations $\mathcal{D}\_n$. The target value $s’$ is determined using the parameters $\alpha$, based on the formula $s’ = s\_{min} - \alpha \times (s\_{max}-s\_{min})$. Here, $s\_{max}$ (worst) and $s\_{min}$ (best) represent the extremities of objective values observed during the search, essentially defining the range of these values.
> Consequently, $s’$ is calculated by adjusting the best observed objective ($s_{min}$) by an amount proportional to the total observed range ($s\_{max} - s\_{min}$). To illustrate, an $\alpha$ value of $-0.1$ corresponds to targeting the 90th percentile of the observed values, while $\alpha = 0.1$ aims to improve upon the best value by the 10th percentile. Conversely, setting $\alpha$ to 0 targets the current best observed value. We set $\alpha=-0.1$ based on observations described in Figure 6 as a value that balanced diversity-performance. We hope this explanation clarifies the role of $\alpha$ in our sampling process.
> * **End-to-end process.** We appreciate this suggestion. The end-to-end procedure iteratively performs three steps: **(1)** sample $M$ candidate points $\\{\tilde{h}\_m\\}\_{m=1}^M $. **(2)** evaluate $M$ points using the surrogate model, i.e. $p(s|\tilde{h}\_m)$ to obtain scores $\\{a(\tilde{h}\_m)\\}\_{m=1}^M$ according to an acquisition function. We use expected improvement (EI), $a(\tilde{h}\_m)= E[\max(p(s|\tilde{h}\_m) - f(h\_{best}), 0)]$. **(3)** select point with the highest score to evaluate next, $h = argmax\_{\tilde{h}\in\\{\tilde{h}\_m\\}\_{m=1}^M}a(\tilde{h})$. We have included these details in App C.
>
> ---
>
> We thank the reviewer for your help in improving our work. Please let us know if our latest changes have addressed your concerns, and if there is anything else you would like to see.

---

> > ### Comment · Reviewer_LVtK · 2023-11-17
> >
> > Revision looks great.  Since my concerns are resolved and my suggestions are reflected, I am increasing the score.

---

> > > ### Author Response · Authors · 2023-11-20
> > > **Thank you**
> > >
> > > Thank you for your constructive comments. We are pleased to have addressed your concerns and appreciate your insights, which have improved the quality of our work.

---

### Official Review · Reviewer_x2Jx · 2023-11-01

**Soundness:** 3 good
**Presentation:** 3 good
**Contribution:** 3 good
**Rating:** 8
**Confidence:** 3

**Summary:**

This paper introduces LLAMBO, an interesting approach that integrates large language models (LLMs) into Bayesian optimization (BO). The authors highlight the challenges of efficiently balancing exploration and exploitation in BO and propose LLAMBO as a solution to enhance various components of model-based BO. LLAMBO leverages the contextual understanding, few-shot learning proficiency, and domain knowledge of LLMs to improve surrogate modeling, candidate sampling, and zero-shot warm-starting. The authors empirically validate LLAMBO's effectiveness in hyperparameter tuning, demonstrating strong performance across diverse benchmarks.

**Strengths:**

1. Generally, this paper is well written and easy to follow.
2. This paper has well shown the feasibility of introducing LLM into BO for further performance improvement through extensive empirical experiments.
3. This paper has conducted extensive experiments to show how and why the performance is improved.

**Weaknesses:**

1. The proposed method may failed to deal with high dimensional optimization problems due to the limited token of LLM.
2. This paper mainly compares with the LLAMBO with standard BO algorithms whereas many neural network-based BO algorithms have been developed recently to improve the modeling of standard BO algorithms, e.g., [R1]. This paper may also compare with it to further support the advantages of using LLM for BO.

[R1] Dai, Z., Shu, Y., Low, B. K. H., & Jaillet, P. (2022). Sample-then-optimize batch neural thompson sampling. Advances in Neural Information Processing Systems, 35, 23331-23344.

**Questions:**

1. In page 8, could the author explain why a negative alpha will help improve the average regret? Intuitively, a negative alpha indicates that the sampling candidates usually perform no better than the best one when compared with the existing candidates, which therefore indicates that no improvement should be made through a negative alpha.
2. What's the dimension of the empirical experiments that have been conducted in this paper? Any high-dimensional ones?

---

> ### Author Response · Authors · 2023-11-16
> **Response to Reviewer x2Jx**
>
> *Thank you for your helpful feedback and comments. We aim to address all the individual points in your review here, but please also see the revised manuscript for changes (highlighted in teal).*
>
> ---
>
> ### [P1] High-dimensional optimization
>
> Thank you for highlighting this concern. Our current work primarily focused on investigating the feasibility of integration of LLMs into BO framework and examining enhancements around various aspects of model-based BO. In this context, our analysis was centered around conventional HPT tasks. These tasks are drawn from widely used benchmarks (e.g. Bayesmark and newly added HPOBench), with dimensionality between $2-8$.
>
> To further bolster our empirical analysis, we included additional results on $24$ tasks from HPOBench, a more recent and more complex HPT benchmark (please see App E.4). We note that in the interest of time, we have only included results from one seeded run, but we plan to include results across runs (which will be updated as they come in). Our analysis reveals LLAMBO achieves strong performance across this extended set of tasks, further validating the efficacy and generalizability of our approach.
>
> With that being said, we agree that it is important to clearly define the scope of our work. Accordingly, we have revised the manuscript in the Introduction and Discussion. We see extending and evaluating high-dimensional BO tasks (e.g. those encountered in neural architecture search) as a promising future step. We see multiple potential avenues to achieve this, including evaluating our current approach with higher-dimensional natural language representations, or applying some search space decomposition techniques.
>
> Regarding the specific limitation related to the LLM’s context window size, our rough calculations indicate that with $25$ iterations, the model we used (gpt-3.5-turbo with $4K$ context window) can support $16$ dimensional problems, which decreases to $8$ dimensions with $50$ iterations. It is noteworthy that the trend in LLM research is moving at pace towards models with larger context windows. Indeed, after the submission of our manuscript, OpenAI has released models featuring context windows ranging from $16K-128K$ ($4\times-32\times$). Therefore, we do not anticipate the context window as a major bottleneck constraining scalability of our approach.
>
> **Actions taken.** We have (1) revised the manuscript to include a discussion of higher-dimensional BO problems and  (2) included additional results on $24$ HPOBench tasks in App D.2 and E.4.
>
> ---
>
> ### [P2] NN baseline
>
> Thank you for pointing out the related work [R1]. This work utilized neural network-based surrogate models to enhance BO’s performance, particularly in high-dimensional problems. Additionally, the method improves computational efficiency by circumventing the need to invert large parameter matrices. In light of your suggestion, we have included a discussion of [R1] in the related works section of the revised manuscript. Furthermore, we empirically compared against it in App E.3. We observed that while STO is competitive against other baselines, especially during early stage of search, LLAMBO is consistently superior and found better solutions more efficiently.
>
> ---
>
> ### [P3] Questions
>
> * Thank you for this question. The use of a low negative $\alpha$ (e.g., $\alpha = -0.1$) aims to target the 'good' regions of the search space, specifically focusing on the top $10$% of observed points. This approach enables the LLM to exploit identified patterns in the existing observations and generate points in proximity to promising solutions. The underlying assumption here is that areas near good solutions are likely to yield similarly high-quality results. This exploitation of the search space could result in identifying candidates with lower regret. We note here that this approach is akin to the sampling strategy of TPE algorithms (which typically uses $\tau=0.25$, which corresponds to constructing density estimators with the top $25$% of observed points). Additionally, this 'good' region dynamically evolves as more points are sampled and evaluated. In our implementation, we set $\alpha=-0.1$ based on observations described in Figure 6 as a value that balanced diversity-performance.
> * For details on the dimensionality of the BO tasks used in our experiments, please refer to App D.
>
> ---
>
> We thank the reviewer for your help in improving our work. Please let us know if our latest changes have addressed your concerns, and if there is anything else you would like to see.

---

> > ### Comment · Reviewer_x2Jx · 2023-11-22
> > **Thank the authors for their detailed response.**
> >
> > I appreciate the clear and detailed response from the authors. Most of my concerns and questions have been addressed. I agree that high-dimensional BO can be a promising direction in the near future especially for LLM-based optimization algorithms and I am looking forward to see that. So, I decided to raise my score.

---

### Official Review · Reviewer_ywEG · 2023-11-07

**Soundness:** 4 excellent
**Presentation:** 4 excellent
**Contribution:** 4 excellent
**Rating:** 8
**Confidence:** 3

**Summary:**

This paper proposes a novel approach for enhancing Bayesian optimization using LLMs. The approach targets several sub problems in BO such as

1) Selecting initial points for warm starting. LLMs being effective at transferring knowledge are can produce a more promising initial set when provided with the problem setup.
2) Surrogate modeling, where the LLM is used to provide prediction and uncertainty estimates for a new design provided past observations.
3) For sampling new candidates to observe.
4) Finally all the steps are augmented for an end-to-end BO approach.

Experimental results on benchmark datasets are promising. Traditional BO work have mainly focussed on black box optimization from scratch, and works on transfer learning are relatively recent. As such the work in this paper is interesting and strongly relevant to the BO community.

**Strengths:**

To summarize, this paper successfully demonstrates the utility of LLMs in Bayesian optimization. The paper makes several strong contributions
- The paper demonstrates the utility of the LLM in all stages of BO from warm starting to candidate selection, and effectively utilizes the power of LLMs in knowledge transfer to novel problems.
- Experimental results show improved performance on several benchmark problems. Experimental evaluation is reasonably extensive including several public and private datasets.

**Weaknesses:**

An obvious weakness of this method is that it has only been evaluated on relatively simple benchmark datasets. The great utilization of this approach would be to select sophisticated neural architectures for novel datasets. However, it is understandable that this may be out of scope for the current work.

**Questions:**

It is not clear why an LLM should have any domain knowledge about hyper-parameter tuning to start with. When provided with past observations as part of the input prompt, it is true that the model may be able to generalize (#). However does the model have any additional knowledge about hyper-parameter tuning as a part of its training data?

Is it obvious that comment (#) is true? What is the mechanism behind the LLM being able to parse numbers and compare them to get a decent understanding of the loss domain?

---

> ### Author Response · Authors · 2023-11-16
> **Response to Reviewer ywEG (1/2)**
>
> *Thank you for your helpful feedback and comments. We aim to address all the individual points in your review here, but please also see the revised manuscript for changes (highlighted in teal).*
>
> ---
>
> ### [P1] Extension to more complex BO tasks
>
> Thank you for highlighting this concern. Our current work primarily focused on *investigating the feasibility* of integration of LLMs into BO framework and examining enhancements around various aspects of model-based BO. In this context, our analysis was centered around conventional HPT tasks. These tasks are drawn from widely used benchmarks (e.g. Bayesmark and newly added HPOBench), with dimensionality between $2-8$.
>
> To further bolster our empirical analysis, we included additional results on $24$ tasks from HPOBench, a more recent and more complex HPT benchmark (please see App E.4). We note that in the interest of time, we have only included results from one seeded run, but we plan to include results across runs (which will be updated as they come in). Our analysis reveals LLAMBO achieves strong performance across this extended set of tasks, further validating the efficacy and generalizability of our approach.
>
> With that being said, we agree that it is important to clearly outline the scope of our work. Accordingly, we have revised the manuscript in the Introduction and Discussion. We see extending and evaluating more complex BO tasks (e.g. those encountered in neural architecture search) as a promising future step. We see multiple potential avenues to achieve this, including evaluating our current approach with higher-dimensional natural language representations, or applying some search space decomposition techniques.
>
> **Actions taken.** We have (1) revised the manuscript to include a discussion of more complex BO problems and (2) included additional results on $24$ HPOBench tasks in App D.2 and E.4.
>
> ---
>
> ### [P2] Prior knowledge about HPT
>
> Thank you for raising this question. The effectiveness of LLAMBO relies on the LLM having useful prior knowledge about correlations in the optimization space. This stems from their extensive training on diverse datasets. It is important to note that due to the Internet-scale data involved in pretraining, it is challenging to pinpoint sources of domain knowledge. However, we conducted various empirical investigations to explore the LLM prior:
> * One of our very first experiments was to verify that LLMs contained knowledge relevant to the optimization problem. This was examined through the warmstarting experiments in (S3). In the **no context** setting, we employed zero-shot prompting, where the LLM was tasked to generate warmstarting points without being provided any dataset-specific information. The goal was to assess the LLM's intrinsic understanding of hyperparameter correlations, independent of specific tasks. We observed that correlation patterns were notably higher than in randomly initialized samples (Figure 2) and warmstarting with these points enhanced the optimization process. This suggests the LLM has knowledge of generalizable correlation patterns between hyperparameters.
> * Furthermore, in our existing analysis of the surrogate model (Section 4.1) and the candidate point sampler (Section 5), we evaluated a specific ablation setting, referred to as **LLAMBO (UnInf)**. This variant operated without detailed problem context or explicit hyperparameter names, relying solely on raw observations. While this setting showed that the LLM could generalize from past observations, we observed improved performance when problem context and hyperparameter names were included. This suggests that the LLM's generalization performance is not solely reliant on past observations but also benefits from prior knowledge.
>
> To address your question directly: while the LLM may/may not have been explicitly trained on hyperparameter tuning datasets, its broad training may allow it to apply general and learned patterns to this domain, which can be particularly effective when combined with specific past observations provided in the prompt.

---

> ### Author Response · Authors · 2023-11-16
> **Response to Reviewer ywEG (2/2)**
>
> ### [P3] Mechanism
>
> We appreciate this question. It is essential to note that the generalization mechanism of ICL is an area of ongoing research and is not fully understood yet. While a detailed exploration of this mechanism is beyond the scope of our current work, we can shed light on some recent advances and hypotheses in the field:
> * **Pattern recognition through ICL.** We leverage the LLM’s ability to recognize patterns in in-context examples and generalize its predictions. Several hypotheses have been put forward to explain ICL: [1] explains it through the lens of implicit Bayesian inference. According to this hypothesis, an LLM, during its pretraining phase, learns to perform Bayesian inference over latent concepts. At the time of inference, the model infers shared latent structures across examples, enabling it to generalize effectively. [2] presents an alternative view, conceptualizing ICL as akin to kernel regression performed on in-context examples. [3] equates in-context learning in attention-based architectures to gradient-based optimization in simplified settings.
> * **ICL applied to optimization.** In our framework, this ability to recognize generalizable patterns from very few examples is used to propose high-potential candidate points (based on discerned patterns of successful solutions) and perform surrogate evaluation of proposed points (by identifying similarities to existing observations). By iterating between processes, we effectively leverage ICL to explore and exploit the loss landscape.
> * **Parsing/interpreting numbers.** Regarding parsing numbers, prior studies [4-5] have demonstrated LLM’s capabilities in reasoning with numerical data. Specifically relevant to our work, recent research [6-7] has explored the use of ICL for numerical prediction tasks in settings such as tabular regression and classification. These studies support our findings that LLMs can indeed discern and leverage numerical patterns effectively, even in few-shot learning scenarios.
>
> [1] Xie, S.M., Raghunathan, A., Liang, P. and Ma, T., 2021, October. An Explanation of In-context Learning as Implicit Bayesian Inference. In International Conference on Learning Representations.
>
> [2] Han, C., Wang, Z., Zhao, H. and Ji, H., 2023. In-Context Learning of Large Language Models Explained as Kernel Regression. arXiv preprint arXiv:2305.12766.
>
> [3] Von Oswald, J., Niklasson, E., Randazzo, E., Sacramento, J., Mordvintsev, A., Zhmoginov, A. and Vladymyrov, M., 2023, July. Transformers learn in-context by gradient descent. In International Conference on Machine Learning (pp. 35151-35174). PMLR.
>
> [4] Imani, S., Du, L. and Shrivastava, H., 2023. Mathprompter: Mathematical reasoning using large language models. arXiv preprint arXiv:2303.05398.
>
> [5] Wallace, E., Wang, Y., Li, S., Singh, S. and Gardner, M., 2019, November. Do NLP Models Know Numbers? Probing Numeracy in Embeddings. In Proceedings of the 2019 Conference on Empirical Methods in Natural Language Processing and the 9th International Joint Conference on Natural Language Processing (EMNLP-IJCNLP) (pp. 5307-5315).
>
> [6] Hegselmann, S., Buendia, A., Lang, H., Agrawal, M., Jiang, X. and Sontag, D., 2023, April. Tabllm: Few-shot classification of tabular data with large language models. In International Conference on Artificial Intelligence and Statistics (pp. 5549-5581). PMLR.
>
> [7] Dinh, T., Zeng, Y., Zhang, R., Lin, Z., Gira, M., Rajput, S., Sohn, J.Y., Papailiopoulos, D. and Lee, K., 2022. Lift: Language-interfaced fine-tuning for non-language machine learning tasks. Advances in Neural Information Processing Systems, 35, pp.11763-11784.
>
> ---
>
> We thank the reviewer for your help in improving our work. Please let us know if our latest changes have addressed your concerns, and if there is anything else you would like to see.

---

### Author Response · Authors · 2023-11-16
**Global Response**

We thank the reviewers for their constructive comments.

We are encouraged that reviewers found LLAMBO novel, as the “first to integrate LLMs into the process of HPO/BO” (**8Jea**), with an approach deemed “interesting and strongly relevant to the BO community” (**ywEG, LVtK**). This includes its role in “combining LLMs with BO” and “knowledge transfer” (**8Jea, ywEG**).

We are pleased that reviewers found our investigations systematic, showcasing a “wealth of ideas” (**8Jea**) and conveying “important messages” (**LVtK**). Additionally, our method was supplemented by “extensive experiments to show how and why the performance is improved” (**x2Jx, ywEG**), resulting in a “strong system overall” (**8Jea**).

We have also taken the reviewers’ feedback into account and made the following key changes to improve the paper:
* Added empirical analysis on $24$ tasks from HPOBench (App E.4), bringing the total number of tasks evaluated to $79$,
* Provided insights on prompt design, including an ablation study on the role of (1) meta-data in the optimization problem description, (2) task-specific instructions (App C.2),
* Integrated a new baseline: a computationally efficient NN surrogate model, STO (App E.3),
* Included 95% CI to all plots and results,
* Moved the related works section into the main paper.

We hope these updates address the reviewers' concerns. We remain open to further feedback.

With thanks,

The Authors of #8133

---

> ### Author Response · Authors · 2023-11-20
> **Brief Update [Nov 20]**
>
> *We wish to thank all reviewers for their insightful reviews and engagement in the rebuttal process.*
>
> We are providing a brief update on our latest revisions. Specifically, we have supplemented the previously partial results with complete findings from the prompt design ablation study (App C2) and HPOBench (App E4).
>
> Kind regards,
>
> The Authors of #8133

---

### Meta-Review · Area_Chair_pxQd · 2023-12-23

**Metareview:**

This work explores the use of LLMs for common AutoML tasks.  Using simple prompts and ICL with GPT-3.5, the authors are able to obtain surprisingly competitive performance on warmstarting (selecting initial designs), prediction, and sampling. Reviewers unanimously gave positive scores to the paper.  The authors show that the use of meta-features ("contexts") can further improve performance.

The paper includes numerous experiments and is easy to follow.  It will certainly stimulate more exploration in the community, as most prior works in this area have focused on compute-intensive training procedures for specialized transformers, while the LLMs here appear to effortlessly provide good results.

- More investigation around the performance of LaMBO. Plugging the warmstarting prompts into GPT 3.5+4.0 I see explanations (and code) that search should be done on a log scale for many standard HPs that should be searched over on a log-scale. The authors make no mention of log-transforming variables, which could explain why random warmstarting and BO perform as poorly as they do.
- More discussion. Some discussion around hypotheses for why the methods work as well as they do would be helpful. For example, beyond reporting on what ChatGPT says about its behavior, one could plot the marginal distributions of HPs selected by LLAMBO, such as learning rates, could help understand how it is focusing the search.  Perhaps GPT has memorized good default values found in code available online? authors may find Perrone et al. (NeurIPS 2019)'s work on learning search spaces (not cited) to be of interest.
- The work exclusively focuses on HPO for ML models, and is given ML-specific information for tasks. The title implies that LLMs can enhance BayesOpt in general, but the entire setup is specific to HPO.  Further, it is unclear how this work is "enhancing" BayesOpt, as it mostly demonstrates how BayesOpt can be replaced with LLMs. I would suggest that the authors find a title that reflects the subject of the paper better, such as 'Large Language Models can do Hyperparameter optimization'.
- It is fishy that GP-DKL outperforms GP for such small dataset sizes.  I have never seen DKL work particularly well in this setting, both in the literature and in practice. Are you sure there isn't a bug in your code? You may want to try using BoTorch via Ax, which will normalize your data appropriately, and can handle the log transformations for initialization and modeling, as suggested above.
- What happens when you run your tasks for more than 30 iterations (e.g., 60 or 100). Nearly all BO papers examine this.
- I had difficulty replicating the results on llama2, codellama.  It is unclear whether this work generalizes to other LLMs, and when we can expect it to work. This minimum level of reproducibility and generalizability should be met in any peer-reviewed publication.

Smaller things:
- Prioritization of content:
  - Prompts. It is difficult to understand the paper without consulting the appendix, particularly wrt the prompts. I would recommend including at least the warmstarting prompts to give readers a sense of what prompts / ICL is being done. Room can be made by cutting non-essential content.
  - Related work. This section is rather short but the authors do not have much space, so I would recommend moving most of this to the appendix, expanding, and only having about one paragraph of review in the MT.
  - Fig2: The correlation plots don't aid in the narrative and so it can be moved to the SM; similarly for generalized variance.
  - MC sampling in surrogate modeling. The MC approach is not an obviously sensible idea (even if (48) tried this) and doesn't add any insights to the paper. Recommend moving this to the SM to make space.
  - Some of the content from Sec B is important for understanding how the sampling works. Please include a minimal description in the MT to make this more comprehensible to readers not familiar with these DRE-type methods.
- The terms 'full', 'partial', 'no context' are not particularly intuitive. In all cases, there is a great deal of contextual information about the task, relative to what is provided to most BO algorithms, so "no context" is a bit confusing.  The AutoML community might identify "partial context" as including "meta-features".
-  LaMBO (Stanton et al. NeurIPS 2022) and (Lin et al. UAI 2021) have similar names. LLMBO might avoid collision.
- Prompts:
  - Fig 10/11: Partial and Full Context prompts are identical except for num instead of number of features. Is this a typo or what partial context means?
  - What is {statistical information} in the warmstarting prompts? Some of this is discussed on page 5, but without any examples it's hard to understand, e.g., what is information about the marginal distributions is given.
  - Some example prompts and answers would be helpful to include. For example, it's not clear exactly what a task descriptor might look like.

**Justification For Why Not Higher Score:**

Work is quite preliminary, and does not sufficiently display its limitations.

**Justification For Why Not Lower Score:**

This work is useful to get out into the community, and already includes quite thorough experiments.

---

### Meta-Review · Program_Chairs · 2024-02-01

**Metareview:**

This paper explores how LLM could help the process of HPO/BO, which is a novel direction. The reviewers are very positive. The AC has raised additional concerns that the authors should take as feedback to further improve this paper. After discussion, the PCs decided to accept this paper.

**Justification For Why Not Higher Score:**

The concerns raised by AC need to be addressed.

**Justification For Why Not Lower Score:**

The topic is novel and interesting and the reviewers are positive.

---

### Decision · Program_Chairs · 2024-01-16

Accept (poster)